# UniEdit-Flow: Unleashing Inversion and Editing in the Era of Flow Models

**Guanlong Jiao**[1,4]**, Biqing Huang**[2]**, Kuan-Chieh Wang**[3]**, Renjie Liao**[1,4,5]

[1]The University of British Columbia    [2]Tsinghua University    [3]Snap Inc.
[4]Vector Institute    [5]Canada CIFAR AI Chair

{gjiao, rjliao}@ece.ubc.ca, hbq@tsinghua.edu.cn, kcjacksonwang@gmail.com

## Abstract

Flow matching models have emerged as a strong alternative to diffusion models, but existing inversion and editing methods designed for diffusion are often ineffective or inapplicable to them. The straight-line, non-crossing trajectories of flow models pose challenges for diffusion-based approaches but also open avenues for novel solutions. In this paper, we introduce a predictor-corrector-based framework for inversion and editing in flow models. First, we propose Uni-Inv, an effective inversion method designed for accurate reconstruction. Building on this, we extend the concept of delayed injection to flow models and introduce Uni-Edit, a region-aware, robust image editing approach. Our methodology is tuning-free, model-agnostic, efficient, and effective, enabling diverse edits while ensuring strong preservation of edit-irrelevant regions. Extensive experiments across various generative models demonstrate the superiority and generalizability of Uni-Inv and Uni-Edit, even under low-cost settings. Project page.

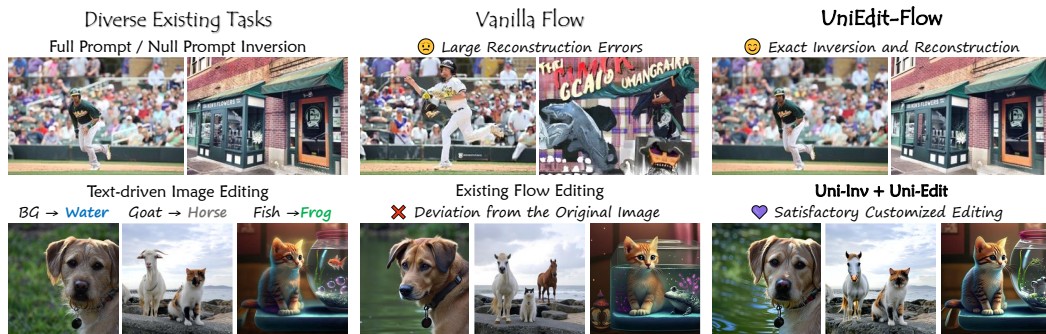

Figure 1: **UniEdit-Flow for image inversion and editing**. Our approach proposes a *highly accurate and efficient*, *model-agnostic*, *training and tuning-free* sampling strategy for flow models to tackle image inversion and editing problems. Cluttered scenes are difficult for inversion and reconstruction, leading to failure results on various methods. Our Uni-Inv achieves **exact reconstruction** even in such complex situations (1st line). Furthermore, existing flow editing always maintain undesirable affects, out region-aware sampling-based Uni-Edit showcases excellent performance for **both editing and background preservation** (2nd line).

# 1 Introduction

Diffusion models have revolutionized the field of image generation and created well-known text-to-image foundation models (Ramesh et al., 2022; Rombach et al., 2022; Ho et al., 2020; Peebles & Xie, 2023). They have also enabled a suite of applications ranging from personalized image generation (Gal et al., 2022; Ruiz et al., 2023; Ye et al., 2023; Wang et al., 2024c), image editing (Avrahami et al., 2022; 2023; Zhang et al., 2023; Bar-Tal et al., 2023), using image models as a prior for 3D generation (Poole et al., 2022; Tang et al., 2023; Yi et al., 2024; Ren et al., 2023), to even non-generative tasks (Sepehri et al., 2024; Yin et al., 2023; Li et al., 2023; Di et al., 2023). Among these, the application of real image editing uniquely leverages the fact that diffusion models learn a mapping between the prior (noise) distribution and the distribution of real images. By simply

adding noise to a real-image, which mimicks the training process, and performing denoising with a new condition (e.g. a modified prompt), a pre-trained diffusion model can be naturally repurposed to an image editor. This has led to a multitude of *inversion* methods (Song et al., 2020a;b; Lu et al., 2022; Wallace et al., 2023; Wang et al., 2025), and a myriad of *training-free, inversion-based* image editing methods (Avrahami et al., 2022; Han et al., 2023; Couairon et al., 2022; Bai et al., 2024; Qian et al., 2024; Brack et al., 2023).

Recently, a new class of models similar to diffusion models known as *flow* models have gained favor and dominated text-to-image tasks such as Stable Diffusion 3 (SD3) (Esser et al., 2024) and Flux (Labs, 2024). These models differ from the the previous generation of models based on diffusion in two key aspects:

1. *Formulation Change* – Flow models are based on deterministic probability flow ordinary differential equations (PF-ODEs), in contrast to diffusion models which are based on SDEs. More specifically, SD3 and Flux uses the rectified flow formulation which models straight lines between the two distributions.

2. *Architecture Change* – In conjunction, there was a shift in architecture from U-Nets with cross-/self-attention layers to using DiTs (Peebles & Xie, 2023) and MM-DiTs (Esser et al., 2024) to improve their *data scaling* ability.

As a result, many methods effective in diffusion models face challenges when applied to flow models. To bridge this gap, recent works have introduced specialized modifications to joint attention in MM-DiTs (Xu et al., 2024b; Avrahami et al., 2024; Wang et al., 2024b; Dalva et al., 2024). Meanwhile, other approaches attempt to reintroduce stochasticity into flow models (Rout et al., 2024; Wang et al., 2024a; Singh & Fischer, 2024), effectively aligning them with diffusion models. However, many of these methods ultimately retrace the trajectory of diffusion models, raising questions about their differences with respect to diffusion and long-term impact.

Our paper aims to re-design inversion and editing by explicitly accounting for the two design changes in the foundation model. We first examine the diffusion-based techniques that fail to transfer effectively to flow models, analyzing their behavior across different architectures. Specifically, we investigate the degradation of the so-called "delayed injection" in flow models, along with the impact of trajectory properties—where vanilla inversion leads to localized sampling errors, as shown in Fig. 1. We argue that the straight-line and non-crossing trajectories of flow models make them prone to accumulating significant errors and even collapsing when velocity estimation is inaccurate during inversion and reconstruction. Furthermore, these properties complicate conditional trajectory guidance, posing challenges for tuning-free editing. Despite these difficulties, we focus on leveraging these characteristics strategically, aiming to unlock the unexplored potential of flow models.

We introduce a novel predictor-corrector-based inversion method for flow models, aiming for accurate and stable reconstruction. Furthermore, we propose a robust sampling-based editing strategy with region-adaptive guidance and velocity fusion, enabling effective and interpretable text-driven image editing. Through both theoretical and empirical analyses, we validate our approach on several benchmarks and demonstrate state-of-the-art performance across diverse generative models, including flow models (Stable Diffusion 3 (Esser et al., 2024) and FLUX (Labs, 2024)) as well as diffusion models (results in Appendix E).

## 2 RELATED WORK

**Inversion.** Modern generative models, particularly diffusion models, aim to map a standard Gaussian distribution to the real data distribution (Goodfellow et al., 2020; Ho et al., 2020). Inversion, the reverse of the generation process, seeks to recover the latent noise corresponding to a given image by reconstructing the diffusion trajectory (Song et al., 2020a; Wallace et al., 2023). The introduction of DDIM (Song et al., 2020a) marked a significant step forward, inspiring a series of high-precision solvers designed to enhance sampling efficiency and minimize inversion errors (Zhang et al., 2024; Lu et al., 2022; Wang et al., 2025; Lu et al., 2022). To further improve alignment between input images and their reconstructions, tuning-based methods have been developed to mitigate reconstruction bias (Mokady et al., 2023; Garibi et al., 2024; Ju et al., 2024; Tumanyan et al., 2023; Yang et al., 2025). More recently, the emergence of flow models has driven the adoption of deterministic samplers, introducing alternative approaches to inversion (Rout et al., 2024; Wang et al., 2024b; Deng

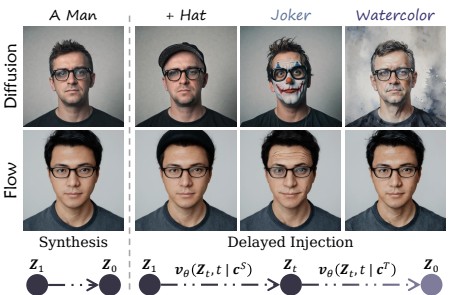

Figure 2: **Delayed injection**, which retains the source condition during the early denoising steps and introduces the edit condition at a middle timestep (illustrated in the bottom part), is a widely used technique in diffusion-based editing (top row). However, when applied to flow models (second row), it is ineffective. While flow-based editing exhibits a mild tendency toward the target edit, it fails to produce sufficiently strong effects.

et al., 2024; Song & Lai, 2024; Singh & Fischer, 2024). These methods design sampling strategies to reduce discretization errors in inversion. However, their non-reconstruction-oriented designs and restrictions on sampler selection limit their applicability to downstream tasks such as image editing. In this work, we focus on the inversion and reconstruction process, designing Uni-Inv to achieve high reconstruction reliability and local exactness, making it well-suited for applications.

**Text-driven Image Editing.** For image editing tasks, early training-based approaches explored generative models to achieve controllable modifications (Zhu et al., 2017; Karras et al., 2019). With the advancement of generative models, the focus has shifted toward training-free editing methods, which offer greater flexibility and efficiency. Tuning-based methods have demonstrated impressive results but require iterative optimization during generation, leading to increased computational costs (Ju et al., 2024; Parmar et al., 2023; Dong et al., 2023; Wu et al., 2023). Meanwhile, attention-manipulation-based techniques leverage multi-branch frameworks for precise control, but their applicability is often restricted to specific model architectures (Cao et al., 2023; Xu et al., 2024a). Sampling-based methods introduce controlled randomness or guidance mechanisms to achieve more flexible editing (Wang et al., 2023; Tsaban & Passos, 2023; Brack et al., 2024; Huberman-Spiegelglas et al., 2024; Kulikov et al., 2024; Mao et al., 2025). More recently, the rise of flow models built on MM-DiTs (Esser et al., 2024) has attracted significant attention in the editing domain due to their strong generative capabilities (Patel et al., 2024; Liu et al., 2023; Sun et al., 2024; Martin et al., 2024; Hu et al., 2024). In this work, we rethink the design of efficient and generalizable image editing methods in the era of flow models, introducing Uni-Edit, a model-agnostic and adaptable approach tailored for text-driven image editing tasks.

## 3 BACKGROUND

### 3.1 FLOW MATCHING

Generative models aim at generating data that follows the real data distribution $\pi_0$ from noise that follows some known distribution $\pi_1$ (*e.g.*, Gaussian distribution). Flow matching (Lipman et al., 2022; Liu et al., 2022; Albergo et al., 2023) proposed to learn a velocity field that is parameterized by a neural network to move noise to data via straight trajectories. The training objective is to solve the following optimization problem:

$$\min_{\theta} \quad \mathbb{E}_{\boldsymbol{Z}_0, \boldsymbol{Z}_1, t} \left[ \|(\boldsymbol{Z}_1 - \boldsymbol{Z}_0) - \boldsymbol{v}_\theta \left( \boldsymbol{Z}_t, t \right)\|^2 \right],$$
$$\boldsymbol{Z}_t = t\boldsymbol{Z}_1 + (1-t)\boldsymbol{Z}_0, \ t \in [0,1], \tag{1}$$

where data $\boldsymbol{Z}_0 \in \pi_0$ and noise $\boldsymbol{Z}_1 \in \pi_1$. $\boldsymbol{Z}_1 - \boldsymbol{Z}_0$ is the target velocity and $\boldsymbol{v}_\theta(\cdot)$ is the learnable velocity field. The trained model is expected to estimate a velocity field to map a randomly sampled Gaussian noise $\boldsymbol{Z}_1 \in \mathcal{N}(0, \boldsymbol{I})$ to generated data $\boldsymbol{Z}_0$ in a deterministic way. This generation process can be viewed as solving an ordinary differentiable equation (ODE) characterized by $\mathrm{d}\boldsymbol{Z}_t = \boldsymbol{v}_\theta \left( \boldsymbol{Z}_t, t \right) \mathrm{d}t$. This ODE can be discretized and solved by solvers such as the Euler method:

$$\boldsymbol{Z}_{t_{i-1}} = \boldsymbol{Z}_{t_i} + (t_{i-1} - t_i) \boldsymbol{v}_\theta \left( \boldsymbol{Z}_{t_i}, t_i \right), \tag{2}$$

where $i \in \{N, \ldots, 0\}$, $t_i$ monotonically increases with $i$, $t_0 = 0$, and $t_N = 1$.

### 3.2 DELAYED INJECTION

Previous works have introduced delayed injection, a simple yet effective technique that helps maintain image consistency during editing. This method preserves the original conditions or reuses the

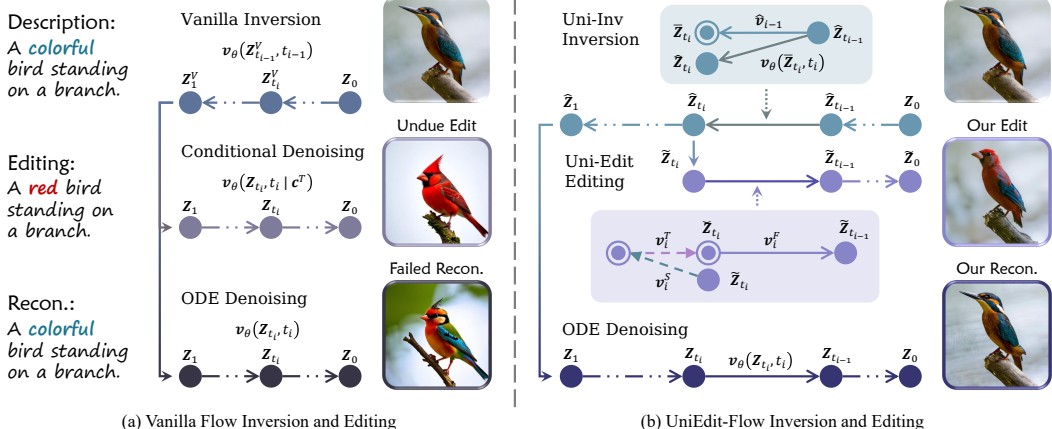

Figure 3: **An overview** of our proposed Uni-Inv and Uni-Edit (bird → red bird). (a) indicates that vanilla flow inversion is incapable for both exact image inversion and controllable editing. (b) demonstrates our proposed Uni-Inv and Uni-Edit, which perform efficient and effective inversion and editing. $\widehat{\boldsymbol{v}}_{i-1}$ indicates the previous velocity. $\boldsymbol{v}_i^S$, $\boldsymbol{v}_i^T$, $\boldsymbol{v}_i^F$ are $\boldsymbol{c}^S$-conditioned, $\boldsymbol{c}^T$-conditioned, and fused velocities. $\boldsymbol{Z}_t$, $\widehat{\boldsymbol{Z}}_t$, $\widetilde{\boldsymbol{Z}}_t$ denote samples of sampling, inversion, and editing, respectively.

latent representations from the inversion process before a specific timestep while injecting the editing conditions afterward, enabling a balanced trade-off between content preservation and targeted modifications (Xiao et al., 2024; Couairon et al., 2022; Wu & De la Torre, 2023). As illustrated in Fig. 2, the top row demonstrates a diffusion-based example, where delayed injection enables an effective trade-off between preserving editing-irrelevant content and achieving the intended editing. Due to the non-linear and intersecting sampling trajectories of diffusion models, modifying conditions midway allows for trajectory transitions, facilitating more flexible and localized image editing (Patel et al., 2024). However, flow models exhibit straight-line and non-intersecting trajectories, which makes it difficult for points on one sampling trajectory to transfer to other trajectories midway (Liu et al., 2022). Such attributes fundamentally hinder the effectiveness of delayed injection, particularly in image editing. As shown in the middle of Fig. 2, applying delayed injection under similar conditions leads to limited improvements in flow-based editing. In this work, we take a deeper look into how to design effective guidance strategies for delayed injection, aiming to unlock more controllable and reliable flow-based image editing.

## 4 METHOD

Fig. 3 provides a brief illustration of image inversion, reconstruction, and editing based on vanilla ODE sampling methods, as well as an overview of our approach. In Fig. 3 (a), due to the mismatch between $\bar{\boldsymbol{Z}}$ and $t$ used in each corresponding forward step, it's difficult for direct inversion to ensure consistency between the reconstructed image and the original one. Besides, conditional denoising without proper guidance cannot enable controllable image editing and leads to undesirable results. In Fig. 3 (b), we implement the idea of correction in two distinct forms for inversion and editing, respectively. In this section, we will present our proposed Uni-Inv and Uni-Edit with the technical contributions for inversion and editing, respectively.

### 4.1 UNI-INV

The motivation of Uni-Inv is to conduct an accurate inversion capable of inverting ODE solutions back to the initial value for particular deterministic samplers. We take the flow model (Liu et al., 2022; Esser et al., 2024) with an Euler method solver as a simple instance of deterministic iterative generation methods facilitated by its concise formula. Denote $\widehat{\boldsymbol{Z}}_t$ as the latent in the inversion process. Eq. (2) describes one iteration step in which we estimate $\boldsymbol{Z}_{t_{i-1}}$ from $\boldsymbol{Z}_{t_i}$ by a denoising step. Through this formulation, given the initial value $\widehat{\boldsymbol{Z}}_0 = \boldsymbol{Z}_0$, the exact value of the inverted latent $\widehat{\boldsymbol{Z}}_{t_i}$ can be derived by the implicit Euler method:

$$\widehat{\boldsymbol{Z}}_{t_i} = \widehat{\boldsymbol{Z}}_{t_{i-1}} - (t_{i-1} - t_i)\, \boldsymbol{v}_\theta(\widehat{\boldsymbol{Z}}_{t_i}, t_i). \tag{3}$$

However, since there is no access to $\widehat{\boldsymbol{Z}}_{t_i}$ in the inversion process, $\boldsymbol{v}_\theta(\widehat{\boldsymbol{Z}}_{t_i}, t_i)$ in Eq. (3) is unknown. Previous tuning-free approaches replace $\boldsymbol{v}_\theta(\widehat{\boldsymbol{Z}}_{t_i}, t_i)$ with an approximation, *e.g.*, $\boldsymbol{v}_\theta(\widehat{\boldsymbol{Z}}_{t_{i-1}}, t_{i-1})$ in DDIM Inversion (Song et al., 2020a;b). Such an approximation assumes that model predictions are consistent across timesteps, which is bound to have errors. Empirically, we find that evaluating $\boldsymbol{v}_\theta(\widehat{\boldsymbol{Z}}_{t_{i-1}}, \cdot)$ using $t_i$, which is similar to the implicit Euler method, instead of $t_{i-1}$, which is adopted in DDIM Inversion, yields more accurate inversion results. This is because eliminating the error of $t$ in function $\boldsymbol{v}_\theta$ can lower the error bound between inversion and sampling. As shown in Fig. 4, using $\boldsymbol{v}_\theta(\widehat{\boldsymbol{Z}}_{t_{i-1}}, t_{i-1})$ (◆) constantly achieves a larger local error compared to $\boldsymbol{v}_\theta(\widehat{\boldsymbol{Z}}_{t_{i-1}}, t_i)$ (■), and ultimately it reconstructs the noise with mismatched background. Nevertheless, the error accumulation of both strategies is non-trivial, as inaccurate velocity estimates continually deviate from the original trajectory. Thus, we propose that the key of inversion for gaining accurate reconstruction lies in finding an approximation $\bar{\boldsymbol{Z}}_{t_i}$ of $\widehat{\boldsymbol{Z}}_{t_i}$ via $\widehat{\boldsymbol{Z}}_{t_{i-1}}$ to align the velocity of Inversion = Reconstruction$^{-1}$.

To estimate a proper $\bar{\boldsymbol{Z}}_{t_i}$, methods like ReNoise (Garibi et al., 2024) suggest utilizing recursive sampling, but this approach significantly increases computational cost. Inspired by the straight trajectories of Rectified Flow (Liu et al., 2022), we propose reusing the previously obtained velocity $\widehat{\boldsymbol{v}}_{i-1}$ at the current time step $t_i$ to push the previous sample $\widehat{\boldsymbol{Z}}_{t_{i-1}}$ as $\bar{\boldsymbol{Z}}_{t_i}$ for the implicit Euler evaluation while avoiding extra model forward. We first use the previous velocity to backtrack the sample from $t_{i-1}$ to $t_i$ as a correction, then obtain the velocity $\widehat{\boldsymbol{v}}_i$ using $\bar{\boldsymbol{Z}}_{t_i}$, which is better aligned with the current timestep $t_i$.

Alg. 1 provides an overview of Uni-Inv. Intuitively, Uni-Inv introduces a correction procedure before performing the inversion step. It first transitions to the high-noise step and estimates the velocity by simulating a denoising procedure. Then it returns to the original low-noise step and performs inversion using the latest "denoising-like" velocity, which can be seen as a closed-form solution of the implicit Euler method. Fig. 4 also provides the error accumulation curves of Uni-Inv (◆), showcasing its superior accuracy. We have the following proposition that bounds the local error, *i.e.*, the one-step error, of Uni-Inv for inversion and reconstruction. It gives the theoretical justification for the high quality of Uni-Inv for reconstruction. The proof is in Appendix A.

**Proposition 4.1.** *Suppose the velocity field $\boldsymbol{v}_\theta$ is Lipschitz, and there is a constant $C$ such that $\left\| \boldsymbol{Z}_{t_p} - \boldsymbol{Z}_{t_q} \right\| \leq C \left\| t_p - t_q \right\|, \forall t_p, t_q \in [0, 1]$, where $\boldsymbol{Z}_{t_p}$ and $\boldsymbol{Z}_{t_q}$ come from the same sampling process. Then for any two consecutive steps $t_{i-1}$ and $t_i$, the local error of inversion and reconstruction using Uni-Inv is $\mathcal{O}\left(\Delta t_i^3\right)$, where $\Delta t_i = t_i - t_{i-1}$.*

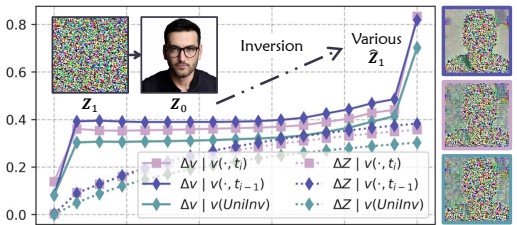

Figure 4: **Per-step error of the velocities and samples** of vanilla inversions. We first synthesis an image $\boldsymbol{Z}_0$ from random noise $\boldsymbol{Z}_1$, then conduct inversions to get various inverted noises $\widehat{\boldsymbol{Z}}_1$ with velocity of $\boldsymbol{v}_\theta(\widehat{\boldsymbol{Z}}_{t_{i-1}}, t_{i-1})$ (◆), $\boldsymbol{v}_\theta(\widehat{\boldsymbol{Z}}_{t_{i-1}}, t_i)$ (■), and Uni-Inv (◆), respectively. We plot the per-step local error of samples ($\Delta \boldsymbol{Z}$) velocities ($\Delta \boldsymbol{v}$). The right shows various inverted $\widehat{\boldsymbol{Z}}_1$, while their border colors correspond to different inversion methods.

---

**Algorithm 1** Uni-Inv (Euler)

**Input:**
Velocity Function $\boldsymbol{v}_\theta$, Initial Image $\boldsymbol{Z}_0 \sim \pi_0$,
Time Steps $t = \{t_0, \ldots, t_N\}$, $t_0 = 0$, $t_N = 1$.

**Initial:**
$\widehat{\boldsymbol{v}}_0 \leftarrow \boldsymbol{v}_\theta(\boldsymbol{Z}_0, t_0)$
$\widehat{\boldsymbol{Z}}_{t_0} \leftarrow \boldsymbol{Z}_0$

**For** $i = 1$ **to** $N$ **do**
1: $\bar{\boldsymbol{Z}}_{t_i} \leftarrow \widehat{\boldsymbol{Z}}_{t_{i-1}} - (t_{i-1} - t_i)\widehat{\boldsymbol{v}}_{i-1}$
2: $\widehat{\boldsymbol{v}}_i \leftarrow \boldsymbol{v}_\theta(\bar{\boldsymbol{Z}}_{t_i}, t_i)$
3: $\widehat{\boldsymbol{Z}}_{t_i} \leftarrow \widehat{\boldsymbol{Z}}_{t_{i-1}} - (t_{i-1} - t_i)\widehat{\boldsymbol{v}}_i$

**End for**
**Output:** $\widehat{\boldsymbol{Z}}_1$

---

### 4.2 UNI-EDIT

Sec. 3.2 provides a case study that highlights the challenges faced by flow models in image editing tasks. As shown in Fig. 5, delayed injection aims to establish an intermediate state between the original and editing trajectories. However, due to the non-intersecting nature of flow trajectories, it is difficult to obtain significantly different directions from middle steps, often resulting in inchoate

**results**. On the other hand, unrestricted direct sampling with editing conditions often leads to **undesirable edit**, as the generated sample diverges from the original trajectory from the outset.

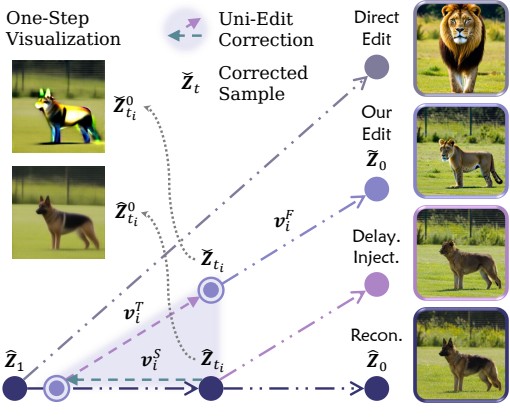

Figure 5: **Demonstration of various sampling-based image editing methods** (dog → lion). Directly utilizing $c^T$ as condition leads to an undue editing. Leveraging delayed injection, which is widely used in diffusion-based methods, inevitably results in an inchoate performance when using deterministic models. Our Uni-Edit mitigates early steps obtained components that are not conducive to editing, ultimately achieving satisfying results.

To address these issues, an intuitive strategy is to inject editing conditions earlier while mitigating excessive modifications using information from the current latent state. Instead of simply injecting conditions earlier, we propose additional correction steps during the editing procedure solely based on the current latent $\widetilde{\boldsymbol{Z}}_{t_i}$. At the injection step, this variable is initialized as the inversion latent, *i.e.*, $\widetilde{\boldsymbol{Z}}_{t_{\alpha N}} = \widehat{\boldsymbol{Z}}_{t_{\alpha N}}$, where $\alpha$ denotes the delay rate and $N$ is the total number of sampling steps. As shown in Fig. 3 and 5, given a source condition $c^S$ that describes the source image and a target condition $c^T$ that specifies the editing objective, we compute two different velocity estimates, $\boldsymbol{v}_i^S = \boldsymbol{v}_\theta(\widetilde{\boldsymbol{Z}}_{t_i}, t_i \mid c^S)$ and $\boldsymbol{v}_i^T = \boldsymbol{v}_\theta(\widetilde{\boldsymbol{Z}}_{t_i}, t_i \mid c^T)$, via the velocity field $\boldsymbol{v}_\theta$. To introduce a correction step, $\widetilde{\boldsymbol{Z}}_{t_i}$ is first transitioned to a previous step with higher noise along the direction of $\boldsymbol{v}_i^S$. It is then mapped back to $\check{\boldsymbol{Z}}_{t_i}$ via $\boldsymbol{v}_i^T$, which is aligned with the current timestep $t_i$. Thus, the sample $\widetilde{\boldsymbol{Z}}_{t_i}$ is corrected by the correction step $\boldsymbol{s}_i \sim (t_{i-1} - t_i)(\boldsymbol{v}_i^T - \boldsymbol{v}_i^S)$ to an edit-friendly stage $\check{\boldsymbol{Z}}_{t_i} = \widetilde{\boldsymbol{Z}}_{t_i} + \boldsymbol{s}_i$, as the visualization in Fig. 5. This procedure corrects undesirable components introduced in early sampling steps that may hinder effective editing. Then, we apply $\boldsymbol{v}_i^F$ (introduced in next part) to move from $\check{\boldsymbol{Z}}_{t_i}$ to $\check{\boldsymbol{Z}}_{t_{i-1}}$, and finally achieving a **proper edit**.

## 4.3 REGION-ADAPTIVE GUIDANCE AND VELOCITY FUSION

To further precisely correct concepts that need to be edited while avoiding excessive damage to the background, a simple idea is to use a mask to determine the edit-relevant regions. Previous works have observed that the difference between latents conditioned on different prompts highlights regions crucial for editing (Couairon et al., 2022; Han et al., 2024). Building on this insight, we leverage this difference $\boldsymbol{v}_i^-$ to construct a mask $\boldsymbol{m}_i = \texttt{MASK}(\boldsymbol{v}_i^-)$, which serves as regional guidance for correction and velocity prediction, thereby improving the controllability of Uni-Edit. Here, $\texttt{MASK}(\cdot)$ denotes the min-max normalization of the channel-wise mean map. To guide the correction step, we first apply a weighting factor of $(1 + \boldsymbol{m}_i)$, encouraging edit-relevant regions to backtrack with a larger stride, thereby enhancing the removal of original concepts crucial for the intended modification. The region guided correction step is $\boldsymbol{s}_i = \omega(t_{i-1} - t_i)(1 + \boldsymbol{m}_i) \odot \boldsymbol{v}_i^-$, where $\omega$ indicates the guidance strength. For the

---

**Algorithm 2** Uni-Edit (Euler)

**Input:**

Velocity Function $\boldsymbol{v}_\theta$, Initial Image $\boldsymbol{Z}_0 \sim \pi_0$,
Source Condition $c^S$, Target Condition $c^T$,
Guidance Strength $\omega$, Delay Rate $\alpha$,
Time Steps $t = \{t_0, \ldots, t_{\alpha N}\}$, $t_0 = 0$, $t_{\alpha N} \leq 1$.

**Initial:**

$\widehat{\boldsymbol{Z}}_{t_{\alpha N}} \leftarrow \text{Uni-Inv}(\boldsymbol{v}_\theta, \boldsymbol{Z}_0, t)$
$\widetilde{\boldsymbol{Z}}_{t_{\alpha N}} \leftarrow \widehat{\boldsymbol{Z}}_{t_{\alpha N}}$

**For** $i = \alpha N$ **to** $1$ **do**

1: $\boldsymbol{v}_i^S, \boldsymbol{v}_i^T \leftarrow \boldsymbol{v}_\theta(\widetilde{\boldsymbol{Z}}_{t_i}, t_i \mid c^S), \boldsymbol{v}_\theta(\widetilde{\boldsymbol{Z}}_{t_i}, t_i \mid c^T)$
2: $\boldsymbol{v}_i^- \leftarrow \boldsymbol{v}_i^T - \boldsymbol{v}_i^S$
3: $\boldsymbol{m}_i \leftarrow \texttt{Mask}(\boldsymbol{v}_i^-)$
4: $\boldsymbol{s}_i \leftarrow \omega(t_{i-1} - t_i)(1 + \boldsymbol{m}_i) \odot \boldsymbol{v}_i^-$
5: $\check{\boldsymbol{Z}}_{t_i} \leftarrow \widetilde{\boldsymbol{Z}}_{t_i} + \boldsymbol{s}_i$
6: $\boldsymbol{v}_i^F \leftarrow \boldsymbol{m}_i \odot \boldsymbol{v}_i^T + (1 - \boldsymbol{m}_i) \odot \boldsymbol{v}_i^S$
7: $\widetilde{\boldsymbol{Z}}_{t_{i-1}} \leftarrow \check{\boldsymbol{Z}}_{t_i} + (t_{i-1} - t_i)\boldsymbol{v}_i^F$

**End for**

**Output:** $\widetilde{\boldsymbol{Z}}_0$

---

subsequent sample update, we fuse the target and source velocities using $\boldsymbol{m}_i$ and $(1 - \boldsymbol{m}_i)$ as respective weights, thus forming the velocity fusion: $\boldsymbol{v}_i^F = \boldsymbol{m}_i \odot \boldsymbol{v}_i^T + (1 - \boldsymbol{m}_i) \odot \boldsymbol{v}_i^S$. Then the sample is updated by $\widetilde{\boldsymbol{Z}}_{t_{i-1}} = \check{\boldsymbol{Z}}_{t_i} + (t_{i-1} - t_i)\boldsymbol{v}_i^F$. The complete editing procedure is outlined in Alg. 2. The delayed injection framework, parameterized by the delay rate $\alpha$, strikes a balance

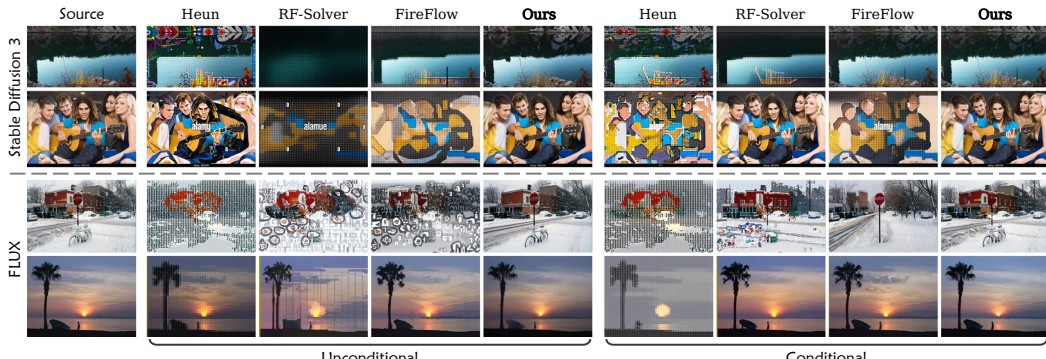

Figure 6: **Qualitative comparison on inversion & reconstruction**. Our method ensures stable reconstruction results in both situations with description accessible (conditional) and unaccessible (unconditional), while taking into account both overall and detail consistency.

between preserving background details and achieving effective modifications, while simultaneously reducing inference costs. By integrating regionally enhanced guidance with velocity fusion, we ultimately obtain an adaptive and computationally efficient editing approach. Furthermore, our velocity fusion method offers advantages over existing latent fusion techniques (Couairon et al., 2022; Han et al., 2024) by providing better performance without additional memory overhead, as illustrated in Fig. 8. We further provide the theoretical analysis of Uni-Edit in Appendix B and details of our model variants in Appendix D.

## 5 EXPERIMENTS

### 5.1 SETUP

**Baselines**: We conduct experiments on two tasks: 1) image inversion and reconstruction; 2) text-driven image editing. For image inversion and reconstruction, we compare our Uni-Inv with the Euler and the Heun method, as well as flow-based methods like RF-Solver (Wang et al., 2024b) and FireFlow (Deng et al., 2024). For text-driven image editing, we compare our Uni-Edit with diffusion-based methods: P2P (Hertz et al., 2022), PnP (Tumanyan et al., 2023), PnP-Inversion (Ju et al., 2024), EditFriendly (Huberman-Spiegelglas et al., 2024), MasaCtrl (Cao et al., 2023), and InfEdit (Xu et al., 2024a), along with the aforementioned two flow-based methods. More tuning-based (Mokady et al., 2023; Garibi et al., 2024) and training-based methods (Wu et al., 2024; Brooks et al., 2023; Shi et al., 2024; Wei et al., 2024) are discussed in Appendix E.

**Benchmarks and Metrics**: For inversion and reconstruction, we report the average MSE, PSNR (Huynh-Thu & Ghanbari, 2008), SSIM (Wang et al., 2004), and LPIPS (Zhang et al., 2018) of reconstructed images on the Conceptual Captions validation dataset (Sharma et al., 2018), which consists of 13.4K images annotated with captions. These metrics are evaluated in both conditional (using image captions as prompts) and unconditional (using null text only) settings. For text-driven image editing, we conduct experiments on PIE-Bench (Ju et al., 2024), which contains 700 images with 10 different editing types. To evaluate edit-irrelevant context preservation, we use structure distance (Tumanyan et al., 2022), along with PSNR and SSIM for annotated unedited regions. The performance of the edits is assessed using CLIP similarity (Radford et al., 2021) for both the whole image and the edited regions. More ablations and results are provided in the Appendix C, D, and H.

**Implementation**: We primarily conduct experiments using `stable-diffusion-3-medium` (SD3) (Esser et al., 2024) and `FLUX.1-dev` (Labs, 2024) models. For inversion and reconstruction, we set the sampling step to 50 for SD3 and 30 for FLUX, while for image editing, we use 15 steps with a delay rate $\alpha$ of 0.6 or 0.8. The relationship between the delay rate and NFE is NFE $= 3\alpha N + 1$. The guidance strength $\omega$ is fixed at 5 for all experiments. Additional results of our method applied to diffusion models (Podell et al., 2023) and various datasets (Hui et al., 2024; Zhao et al., 2024) are provided in Appendix I.

Table 1: **Quantitative results for inversion and reconstruction**. For Stable Diffusion 3 (SD3) (Esser et al., 2024), we keep each method's NFE close to 100, which means we set sampling step to 50 for once-forward methods (*i.e.*, Euler, FireFlow, and Ours) and to 25 for twice-forward methods (*i.e.*, Heun and RF-Solver). For FLUX (Labs, 2024), we keep NFE close to 60 (*i.e.*, 30 for once-forward methods and 15 steps for twice-forward methods). We adopt the official implementations of baselines for FLUX, and reimplement their methods for SD3. The best and second best results are **bolded** and underlined, respectively. Cells are highlighted from worse to better .

| Method | Model | Unconditional | | | | Conditional | | | |
|---|---|---|---|---|---|---|---|---|---|
| | | $MSE^{\downarrow}_{10^3}$ | $PSNR^{\uparrow}$ | $SSIM^{\uparrow}_{10^2}$ | $LPIPS^{\downarrow}_{10^2}$ | $MSE^{\downarrow}_{10^3}$ | $PSNR^{\uparrow}$ | $SSIM^{\uparrow}_{10^2}$ | $LPIPS^{\downarrow}_{10^2}$ |
| Euler | SD3 | 29.98 | 16.54 | 58.97 | 29.03 | 29.14 | 16.80 | 57.54 | 29.92 |
| Heun | | 25.34 | 16.98 | 67.25 | 26.63 | 26.32 | 16.89 | 64.14 | 27.70 |
| RF-Solver | | 45.24 | 15.43 | 56.57 | 33.67 | 26.54 | 18.52 | 64.10 | 26.38 |
| FireFlow | | 20.27 | 19.60 | 66.96 | 25.29 | 16.95 | 20.85 | 68.89 | 22.83 |
| **Ours** | | **11.52** | **21.81** | **78.89** | **12.85** | **7.86** | **23.41** | **82.23** | **9.53** |
| Euler | FLUX | 90.59 | 11.10 | 40.51 | 43.13 | 85.69 | 11.43 | 37.64 | 44.31 |
| Heun | | 83.04 | 11.77 | 42.10 | 39.96 | 76.79 | 12.17 | 40.17 | 41.18 |
| RF-Solver | | 29.32 | 16.97 | 57.17 | 31.75 | 34.71 | 16.38 | 55.64 | 32.43 |
| FireFlow | | 23.31 | 18.15 | 63.85 | 27.96 | 30.78 | 17.59 | 62.51 | 28.30 |
| **Ours** | | **8.85** | **22.15** | **79.45** | **17.10** | **14.36** | **20.91** | **77.09** | **20.27** |

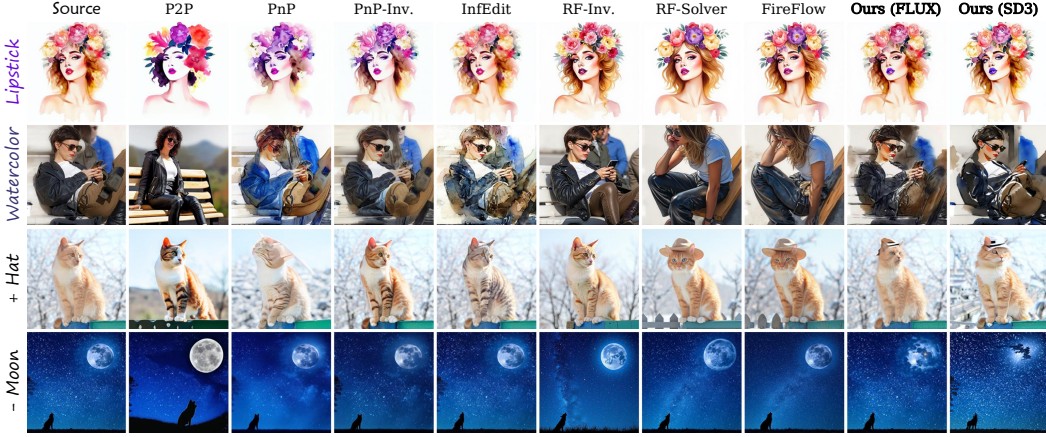

Figure 7: **Qualitative comparison on image editing**. Our method consistently achieves more appropriate editing with better background preservation across various flow models.

## 5.2 REAL IMAGE INVERSION & RECONSTRUCTION

**Quantitative Comparison**: Tab. 1 provides the quantitative results for reconstruction across various flow-based methods. We reconstruct images using only the inverted noise, without utilizing latent features from the inversion process. The results demonstrate that our proposed Uni-Inv consistently outperforms the baselines across different models and settings, including the unconditional case where text description is absent.

**Qualitative Comparison**: Fig. 6 shows the qualitative comparison of inversion and reconstruction across methods. Our method achieves nearly identical reconstruction results in both unconditional and conditional settings for various flow models. In contrast, other methods struggle with reconstruction without text conditions and show weaker performance even when image captions are available. These results strongly highlight the effectiveness of our Uni-Inv.

## 5.3 TEXT-DRIVEN IMAGE EDITING

**Quantitative Comparison**: Tab. 2 presents the quantitative results for text-driven image editing. Our method, Uni-Edit, consistently outperforms other approaches, particularly excelling in CLIP

Table 2: **Text-driven image editing comparison** on PIE-Bench (Ju et al., 2024). We report the peer-reviewed results of each baseline, and evaluate our proposed Uni-Edit using the relatively lightweight Stable Diffusion 3 (SD3) (Esser et al., 2024) and FLUX (Labs, 2024) to demonstrate the effectiveness. We set $\omega = 5$ and mark (N, $\alpha$) in the subscript. The best and second best results are **bolded** and underlined, respectively. Cells are highlighted from worse to better .

| Method | Model | Struc. Dist.$^{\downarrow}_{10^3}$ | BG Preservation PSNR↑ | LPIPS$^{\downarrow}_{10^3}$ | MSE$^{\downarrow}_{10^4}$ | SSIM$^{\uparrow}_{10^2}$ | CLIP Sim.↑ Whole | Edited | Steps | NFE |
|---|---|---|---|---|---|---|---|---|---|---|
| P2P | Diff. | 69.43 | 17.87 | 208.80 | 219.88 | 71.14 | 25.01 | 22.44 | 50 | 100 |
| PnP | Diff. | 28.22 | 22.28 | 113.46 | 83.64 | 79.05 | 25.41 | 22.55 | 50 | 100 |
| PnP-Inv. | Diff. | 24.29 | 22.46 | 106.06 | 80.45 | 79.68 | 25.41 | 22.62 | 50 | 100 |
| EditFriendly | Diff. | - | 24.55 | 91.88 | 95.58 | 81.57 | 23.97 | 21.03 | 50 | 100 |
| MasaCtrl | Diff. | 28.38 | 22.17 | 106.62 | 86.97 | 79.67 | 23.96 | 21.16 | 50 | 100 |
| InfEdit | Diff. | 13.78 | 28.51 | **47.58** | 32.09 | 85.66 | 25.03 | 22.22 | 12 | 72 |
| RF-Inv. | FLUX | 40.60 | 20.82 | 159.62 | 96.01 | 71.92 | 25.20 | 22.11 | 28 | 56 |
| RF-Solver | FLUX | 31.10 | 22.90 | 146.11 | 80.70 | 81.90 | 26.00 | 22.88 | 15 | 60 |
| FireFlow | FLUX | 28.30 | 23.28 | 130.61 | 71.01 | 82.82 | 25.98 | 22.94 | 15 | 32 |
| **Ours** $_{(15, 0.6)}$ | SD3 | 21.40 | 24.96 | 89.78 | 49.20 | 86.11 | 26.39 | 22.72 | 15 | 28 |
| **Ours** $_{(15, 0.8)}$ | FLUX | 26.85 | 24.10 | 112.71 | 61.30 | 84.86 | **26.97** | **23.51** | 15 | 37 |
| **Ours** $_{(15, 0.6)}$ | FLUX | **10.14** | **29.54** | 64.77 | **18.30** | **90.42** | 25.80 | 22.33 | 15 | 28 |

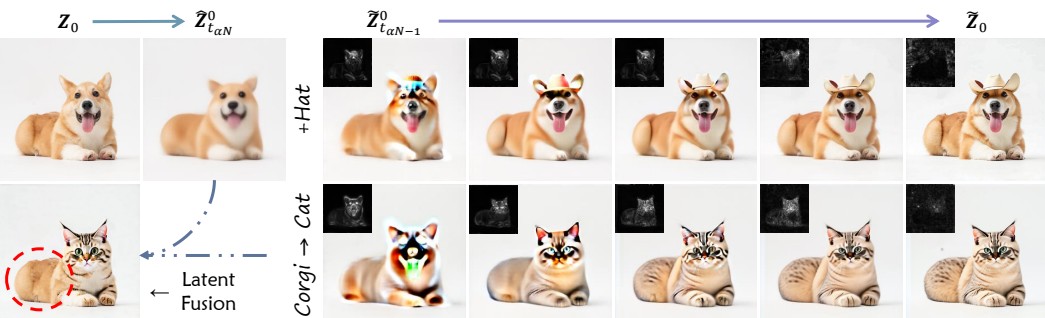

Figure 8: **Visualization of Uni-Edit process**. The guidance mask of each denoising step is shown at the upper right of the image. We also demonstrate the "Sphinx" phenomenon that existing latent fusion approaches may cause at the lower left of the figure.

similarity. Based on SD3, Uni-Edit achieves a balance between background preservation and editing effectiveness, outperforming existing flow-based methods in both areas. On FLUX, it surpasses the inherent limitations of training-free approaches in background preservation while maintaining competitive editing performance. More results on diffusion models are given in Appendix E.

**Qualitative Comparison**: Fig. 7 compares the visual editing outcomes across methods. Attention-based approaches often transfer attributes to unrelated areas (*e.g.*, P2P, PnP, and PnP-Inversion add purple to the overall image rather than focusing on specific regions, like lipstick). Sampling-based methods without regional constraints suffer from mis-edits or insufficient editing (*e.g.*, RF-Solver and FireFlow). In contrast, our method provides precise guidance for both local modification and global stylization. Additional results on various datasets (Zhao et al., 2024; Hui et al., 2024) are provided in Appendix I.

**Procedure Analysis**: Fig. 8 visualizes the editing procedure. The superscript 0 indicates the result is directly transitioned from the sample to $t = 0$ using its current velocity. Early steps focus on broader areas with stronger editing intensity to eliminate original concepts, while later steps refine details, reducing the influence of $\boldsymbol{m}_i$. We also show results from the existing latent fusion method (Couairon et al., 2022; Han et al., 2024), which uses masks to fuse inversion and edit latents. These results lead to unnatural, "Sphinx"-like outputs, highlighting the adaptability and efficiency of our approach. More ablation studies, applications, and video editing results are shown in the Appendix.

## 6 CONCLUSION

We introduce a novel, tuning-free, model-agnostic methodology that combines the reconstruction-effective inversion method Uni-Inv with a region-aware, training-free image editing strategy Uni-Edit. To exploit the properties of flow models, we design robust, region-adaptive guidance in Uni-Edit to enhance the delayed injection framework, supported by Uni-Inv. Extensive experiments validate the effectiveness of our approach, demonstrating remarkable results while maintaining low inference costs. We will explore more diverse conditions (*e.g.*, adopt an image as a personalization prompt) that can be injected for further customized image editing in the future.

## 7 ACKNOWLEDGEMENTS

This work was supported, in part, by the NSERC DG Grant (No. RGPIN-2022-04636), the Vector Institute for AI, the Canada CIFAR AI Chair program, and a gift fund from Snap Inc. Resources used in preparing this research were provided, in part, by the Province of Ontario, the Government of Canada through the Digital Research Alliance of Canada `alliance.can.ca`, and companies sponsoring the Vector Institute `www.vectorinstitute.ai/#partners`, and Advanced Research Computing at the University of British Columbia. Additional resource was provided by the Canada Foundation for Innovation (CFI) via the John R. Evans Leaders Fund (JELF). G.J. is supported by a UBC Four Year Doctoral Fellowship.

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

APPENDIX

# A PROOF OF PROPOSITION 4.1

Prop. 4.1: *Suppose the velocity field $v_\theta$ is Lipschitz, and there is a constant $C$ such that $\left\| Z_{t_p} - Z_{t_q} \right\| \leq C \left\| t_p - t_q \right\|, \forall t_p, t_q \in [0, 1]$, where $Z_{t_p}$ and $Z_{t_q}$ come from the same sampling process. Then for any two consecutive steps $t_{i-1}$ and $t_i$, the local error of inversion and reconstruction using Uni-Inv is $\mathcal{O}\left(\Delta t_i^3\right)$, where $\Delta t_i = t_i - t_{i-1}$.*

**Assumption 1.** *The velocity function $v_\theta(\cdot, \tau)$ is $X_1$-Lipschitz for $\forall \ \tau \in [0, 1]$, i.e., given a $\tau$, $\left\| v_\theta(\zeta_1, \tau) - v_\theta(\zeta_2, \tau) \right\| \leq X_1 \left\| \zeta_1 - \zeta_2 \right\|$ for $\forall \ \zeta_1, \zeta_2$.*

**Assumption 2.** *The velocity function $v_\theta(\zeta, \cdot)$ is $X_2$-Lipschitz for $\forall \ \zeta$, i.e., given a $\zeta$, $\left\| v_\theta(\zeta, \tau_1) - v_\theta(\zeta, \tau_2) \right\| \leq X_2 \left\| \tau_1 - \tau_2 \right\|$ for $\forall \ \tau_1, \tau_2$.*

**Assumption 3.** $\left\| Z_{t_p} - Z_{t_q} \right\| \leq C \left\| t_p - t_q \right\|, \forall \ p, q \in [0, 1]$ *when $Z_{t_p}$ and $Z_{t_q}$ come from the same trajectory.*

*Proof.* Given a deterministic solver, *e.g.* Euler's method:

$$Z_{t_{i-1}} = Z_{t_i} + (t_{i-1} - t_i) \, v_\theta \left( Z_{t_i}, t_i \right). \tag{A.1}$$

The corresponding inversion step of Uni-Inv is denoted by:

$$\widehat{Z}_{t_i} = \widehat{Z}_{t_{i-1}} - (t_{i-1} - t_i) \, \widehat{v}_i, \tag{A.2}$$

where $\widehat{v}_i$ is obtained via Algo. 1 and can be expressed as:

$$\widehat{v}_i = v_\theta \left( \widehat{Z}_{t_{i-1}} - (t_{i-1} - t_i) \, \bar{v}_{i-1}, t_i \right). \tag{A.3}$$

Define the estimation error $\mathcal{E}_i$ as $\mathcal{E}_i = \left\| Z_{t_i} - \widehat{Z}_{t_1} \right\|$. Bringing Eq. A.1 and Eq. A.2 into it, we obtain that:

$$\mathcal{E}_i = (t_{i-1} - t_i) \left\| \widehat{v}_i - v_\theta \left( Z_{t_i}, t_i \right) \right\|. \tag{A.4}$$

Denote $\mathcal{E}_i^1 = \left\| \widehat{v}_i - v_\theta \left( Z_{t_i}, t_i \right) \right\|$, we can bring in Eq. A.3:

$$\mathcal{E}_i^1 = \left\| v_\theta \left( \widehat{Z}_{t_{i-1}} - (t_{i-1} - t_i) \, \bar{v}_{i-1}, t_i \right) - v_\theta \left( Z_{t_i}, t_i \right) \right\|. \tag{A.5}$$

Using the Lipschitz continuity of $v_\theta(\cdot, \tau)$, we have:

$$\mathcal{E}_i^1 \leq X_1 \left\| \widehat{Z}_{t_{i-1}} - (t_{i-1} - t_i) \, \bar{v}_{i-1} - Z_{t_i} \right\|. \tag{A.6}$$

Bring in Eq. A.1 for $Z_{t_i}$, there is:

$$
\begin{aligned}
\mathcal{E}_i^1 &\leq X_1 \left\| \left( \widehat{Z}_{t_{i-1}} - Z_{t_{i-1}} \right) + (t_{i-1} - t_i) \left( v_\theta \left( Z_{t_i}, t_i \right) - \bar{v}_{i-1} \right) \right\| \\
&\leq X_1 \left\| \widehat{Z}_{t_{i-1}} - Z_{t_{i-1}} \right\| + X_1 (t_{i-1} - t_i) \left\| v_\theta \left( Z_{t_i}, t_i \right) - \bar{v}_{i-1} \right\|.
\end{aligned}
\tag{A.7}
$$

The first term is the accumulative error of the previous steps. We denote it as $\mathcal{E}_i^A = \left\| \widehat{Z}_{t_{i-1}} - Z_{t_{i-1}} \right\|$ and it should be neglected for local error analysis. We further denote $\mathcal{E}_i^2 = \left\| v_\theta \left( Z_{t_i}, t_i \right) - \bar{v}_{i-1} \right\|$. To analyse this item, we first consider a second-order case, *i.e.*, utilizing an additional function evaluation step to calculate $\bar{v}_{i-1} = v_\theta \left( \widehat{Z}_{t_{i-1}}, t_{i-1} \right)$. Then we have:

$$\mathcal{E}_i^2 = \left\| v_\theta \left( Z_{t_i}, t_i \right) - v_\theta \left( \widehat{Z}_{t_{i-1}}, t_{i-1} \right) \right\|. \tag{A.8}$$

Using the Lipschitz continuity of $v_\theta(\cdot, \tau)$ and $v_\theta(\zeta, \cdot)$, we have:

$$
\begin{aligned}
\mathcal{E}_i^2 &= \left\| v_\theta \left( Z_{t_i}, t_i \right) - v_\theta \left( Z_{t_i}, t_{i-1} \right) + v_\theta \left( Z_{t_i}, t_{i-1} \right) - v_\theta \left( \widehat{Z}_{t_{i-1}}, t_{i-1} \right) \right\| \\
&\leq \left\| v_\theta \left( Z_{t_i}, t_i \right) - v_\theta \left( Z_{t_i}, t_{i-1} \right) \right\| + \left\| v_\theta \left( Z_{t_i}, t_{i-1} \right) - v_\theta \left( \widehat{Z}_{t_{i-1}}, t_{i-1} \right) \right\| \\
&\leq X_1 \left\| Z_{t_i} - \widehat{Z}_{t_{i-1}} \right\| + X_2 \left\| t_i - t_{i-1} \right\|.
\end{aligned}
\tag{A.9}
$$

We denote $\Delta t_i = t_i - t_{i-1}$. Using the Assumption 3, we have:

$$
\begin{aligned}
\mathcal{E}_i^2 &\leq X_1 \left( \left\| \boldsymbol{Z}_{t_i} - \boldsymbol{Z}_{t_{i-1}} \right\| + \left\| \boldsymbol{Z}_{t_{i-1}} - \widehat{\boldsymbol{Z}}_{t_{i-1}} \right\| \right) + X_2 \Delta t_i \\
&\leq \left( C X_1 + X_2 \right) \Delta t_i + X_1 \mathcal{E}_i^A.
\end{aligned}
\tag{A.10}
$$

Ultimately, the estimation error is as follows:

$$
\begin{aligned}
\mathcal{E}_i &\leq \Delta t_i \mathcal{E}_i^1 \leq \Delta t_i \left( X_1 \mathcal{E}_i^A + X_1 \Delta t_i \mathcal{E}_i^2 \right) \\
&\leq \Delta t_i \left( X_1 \mathcal{E}_i^A + X_1 \Delta t_i \left( \left( C X_1 + X_2 \right) \Delta t_i + X_1 \mathcal{E}_i^A \right) \right) \\
&= X_1 \left( C X_1 + X_2 \right) \Delta t_i^3 + \left( X_1^2 \Delta t_i^2 + X_1 \Delta t_i \right) \mathcal{E}_i^A.
\end{aligned}
\tag{A.11}
$$

For local error analysis, we neglect the global accumulated error $\mathcal{E}_i^A$, then we have the local error $\mathcal{E}_i^L$:

$$
\mathcal{E}_i^L \leq X_1 \left( C X_1 + X_2 \right) \Delta t_i^3 = \mathcal{O} \left( \Delta t_i^3 \right).
\tag{A.12}
$$

Furthermore, since the step count of the iterative algorithm is $\mathcal{O} \left( 1 / \Delta t_i \right)$, we can have the global error: $\mathcal{E}^G = \mathcal{O} \left( \max_i \left( \Delta t_i^2 \right) \right)$.

Now, let's go back to the second-order case assumption $\bar{\boldsymbol{v}}_{i-1} = \boldsymbol{v}_\theta \left( \widehat{\boldsymbol{Z}}_{t_{i-1}}, t_{i-1} \right)$ we mentioned earlier. From a practical perspective, Algo. 1 provides an additional function evaluation in the initialization stage, making its first step standard second-order case. After that, in the ideal case, each $\bar{\boldsymbol{v}}_i$ should converge to $\boldsymbol{v}_i$, and thus the first-order approximation of the algorithm does not significantly affect the error. Theoretically, since that:

$$
\begin{aligned}
\bar{\boldsymbol{v}}_{i-1} &= \boldsymbol{v}_\theta \left( \bar{\boldsymbol{Z}}_{i-1}, t_{i-1} \right), \\
\bar{\boldsymbol{Z}}_{i-1} &= \widehat{\boldsymbol{Z}}_{i-2} - \left( t_{i-2} - t_{i-1} \right) \bar{\boldsymbol{v}}_{i-2},
\end{aligned}
\tag{A.13}
$$

neglecting the last-step accumulated error, we can derive that

$$
\left\| \widehat{\boldsymbol{Z}}_{i-1} - \bar{\boldsymbol{Z}}_{i-1} \right\| \leq \Delta t_{i-1} \left\| \widehat{\boldsymbol{v}}_{i-2} - \bar{\boldsymbol{v}}_{i-2} \right\|.
\tag{A.14}
$$

Using the Lipschitz continuity of $\boldsymbol{v}_\theta(\cdot, \tau)$, we can get:

$$
\left\| \widehat{\boldsymbol{v}}_{i-2} - \bar{\boldsymbol{v}}_{i-2} \right\| \leq X_1 \left\| \widehat{\boldsymbol{Z}}_{i-2} - \bar{\boldsymbol{Z}}_{i-2} \right\|.
\tag{A.15}
$$

Neglecting the last-step accumulated error of velocity estimation for local error calculation, we note that the one-order approximation brings no change to the conclusion of $\mathcal{E}_i^L = \mathcal{O} \left( \Delta t_i^3 \right)$.

This local error serves the dual processes of inversion and reconstruction, theoretically ensuring the effectiveness of our proposed inversion in practical applications. Moreover, our approach does not utilize the derivative approximation to achieve the result, only expects the velocity function to have good mathematical properties.

## B  THEORETICAL ANALYSIS OF UNI-EDIT

Conducting a mathematical perspective, our proposed Uni-Edit can be compressed into a single arithmetic representation. As shown in Algo. 2, we have the editing result:

$$
\begin{aligned}
\widetilde{\boldsymbol{Z}}_{t_{i-1}} &= \check{\boldsymbol{Z}}_{t_i} + \left( t_{i-1} - t_i \right) \boldsymbol{v}_i^F \\
&= \widetilde{\boldsymbol{Z}}_{t_i} + \boldsymbol{s}_i + \left( t_{i-1} - t_i \right) \boldsymbol{v}_i^F \\
&= \widetilde{\boldsymbol{Z}}_{t_i} + \omega \left( t_{i-1} - t_i \right) \left( \mathbf{1} + \boldsymbol{m}_i \right) \odot \left( \boldsymbol{v}_i^T - \boldsymbol{v}_i^S \right) \\
&\quad + \left( t_{i-1} - t_i \right) \left( \boldsymbol{m}_i \odot \boldsymbol{v}_i^T + \left( \mathbf{1} - \boldsymbol{m}_i \right) \odot \boldsymbol{v}_i^S \right) \\
&= \widetilde{\boldsymbol{Z}}_{t_i} + \left( t_{i-1} - t_i \right) \boldsymbol{v}_i^*,
\end{aligned}
\tag{B.16}
$$

and the reformed velocity is:

$$
\begin{aligned}
\boldsymbol{v}_i^* &= \omega \left( \mathbf{1} + \boldsymbol{m}_i \right) \odot \left( \boldsymbol{v}_i^T - \boldsymbol{v}_i^S \right) + \left( \boldsymbol{m}_i \odot \boldsymbol{v}_i^T + \left( \mathbf{1} - \boldsymbol{m}_i \right) \odot \boldsymbol{v}_i^S \right) \\
&= \boldsymbol{v}_i^S + \left( \omega \left( \mathbf{1} + \boldsymbol{m}_i \right) + \boldsymbol{m}_i \right) \odot \left( \boldsymbol{v}_i^T - \boldsymbol{v}_i^S \right),
\end{aligned}
\tag{B.17}
$$

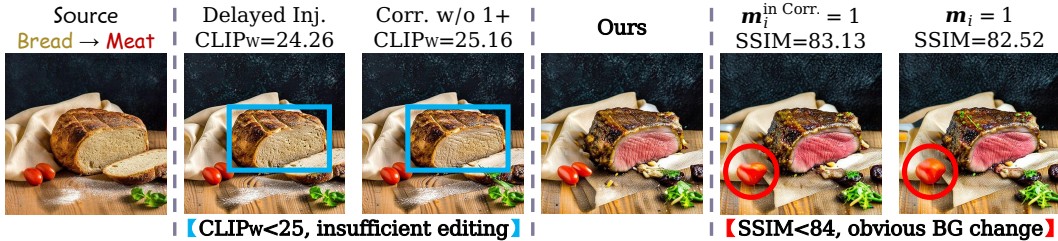

Figure C.1: Illustrations of insufficient editing and background destruction. CLIP$_w$ indicates the whole CLIP similarity.

Table C.1: Component ablation studies of Uni-Edit on PIE-Bench using Stable Diffusion 3. We set step = 15, $\alpha = 0.6$, and $\omega = 5.0$ in these experiments. "w/o Uni-Inv" means using DDIM Inversion-like Euler inversion to replace Uni-Inv in the editing procedure. "w/o Uni-Edit" indicates using naive delayed injection (using $v_i^T$ after inversion) for editing. "Corr." represents the Correction $s_i$ in Uni-Edit. "Corr. w/o 1+" indicates using $m_i$ as the mask weight of $s_i$ instead of $(1 + m_i)$. "$m_i^{\text{in V.F.}} = 1$" means using $1$ to replace the mask $m_i$ in Velocity Fusion $v_i^F$ ($v_i^F = v_i^T$), which can be seen as Uni-Edit without Velocity Fusion. "$m_i^{\text{in V.F.}} = 0$" means using $0$ to replace the mask $m_i$ in $v_i^F$ ($v_i^F = v_i^T$). "$m_i^{\text{in Corr.}} = 1$" indicates using $1$ to replace the mask $m_i$ in the Correction $s_i$, which can be seen as Uni-Edit without Correction. "$m_i = 1$" claims performing editing without region-adaptive guidance, which is equivalent to using simple classifier-free guidance (CFG).

| Method | Structure | Background Preservation | | CLIP Similarity↑ | |
| --- | --- | --- | --- | --- | --- |
| | Distance$^\downarrow_{10^3}$ | PSNR↑ | SSIM$^\uparrow_{10^2}$ | Whole | Edited |
| w/o Uni-Inv | 40.87 | 21.93 (-3.03) | 74.90 (-11.21) | 25.54 (-0.85) | 21.93 (-0.79) |
| w/o Uni-Edit | 9.78 | 27.92 (+2.96) | 89.62 (+3.51) | 24.26 (-2.13) | 20.92 (-1.80) |
| w/o Corr. | 9.45 | 28.00 (+3.04) | 89.67 (+3.56) | 23.78 (-2.61) | 20.52 (-2.20) |
| Inv. $v(\cdot, t_i)$ | 36.95 | 23.82 (-1.14) | 80.25 (-5.86) | 25.99 (-0.40) | 22.08 (-0.64) |
| Corr. w/o 1+ | 11.33 | 27.44 (+2.48) | 89.31 (+3.20) | 25.16 (-1.23) | 21.72 (-1.00) |
| $m_i^{\text{in V.F.}} = 1$ | 22.92 | 24.62 (-0.34) | 85.50 (-0.61) | 26.39 (-0.00) | 22.74 (+0.02) |
| $m_i^{\text{in V.F.}} = 0$ | 21.78 | 25.01 (+0.05) | 86.13 (+0.02) | 26.22 (-0.18) | 22.57 (-0.15) |
| $m_i^{\text{in Corr.}} = 1$ | 28.98 | 23.36 (-1.60) | 83.13 (-2.98) | 26.55 (+0.16) | 22.80 (+0.08) |
| $m_i = 1$ | 30.48 | 23.07 (-1.89) | 82.52 (-3.59) | 26.53 (+0.14) | 22.83 (+0.11) |
| **Ours** | 21.40 | 24.96 | 86.11 | 26.39 | 22.72 |

It's interesting that we finally obtain a velocity $v_i^*$ which is very similar to the classifier-free guidance (CFG) (Chung et al., 2022) but with a per-pixel-variant weight instead of a single constant value. Some previous works consider CFG as a predictor-corrector (Song et al., 2020a; Bradley & Nakkiran, 2024). From this perspective, whereas we take a different yet more interpretable approach to conduct a predictor-corrector, and eventually obtain a method with adaptive guidance strength for different regions. In the manuscript, we experimentally validate that the mask obtained from our designed sampling strategy is rationally adaptive to vary with the editing objective and the iteration step. Therefore, our method ensures to achieve per-pixel adaptive guidance strength within the framework of predictor-corrector, which in turn confers effectiveness for text-driven image editing to flow models.

## C ABLATION STUDIES OF UNI-EDIT

### C.1 COMPONENT ABLATIONS

Preliminary, as shown in Fig. C.2, text-driven image editing tasks pursue a trade-off between editing effect and background preservation. The result we hope for is to preserve non-editing regions while also achieving the editing requirements of the image. Empirically, on PIE-bench (Ju et al., 2024),

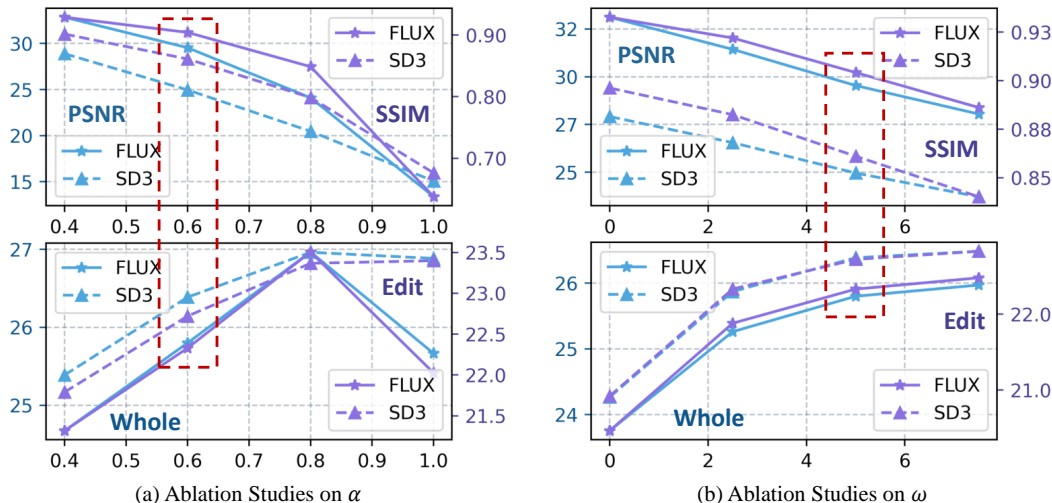

Figure C.2: **Ablation studies of Uni-Edit** on (a) delay rate $\alpha$ and (b) guidance strength $\omega$. The top indicates the background preservation, while the bottom refers to CLIP scores of editing.

$\text{CLIP}_w < 25$ always means the editing impact is insignificant (as the blue box in Fig. C.2), while $\text{SSIM} < 84$ usually indicates the background is destructed (as the red circle in Fig. C.2).

We provide ablation studies of the main components of Uni-Edit in Tab. C.1, discussing the impacts of Uni-Inv, Uni-Edit, Correction, Velocity Fusion, and the mask in image editing tasks. Without Uni-Inv (w/o Uni-Inv), the background preservation decreases significantly, indicating that inverted noisy latent, which is capable of accurate reconstruction, is necessary for controllable editing. The results of naive delayed injection (w/o Uni-Edit) show the importance of well-designed guidance for flow-based image editing, which is just as discussed in the manuscript. Meanwhile, the Correction provides guidance targeted to the editing objective, thus unleashing image editing of flow models. When disabling the Correction (w/o Corr.), the result shows almost no editing. Regarding the mask $\boldsymbol{m}_i$, we can demonstrate through relative experiments that it plays an important role in the trade-off between background preservation and editing effect. The mask $\boldsymbol{m}_i$ enhances the correction and editing strength of editing-related regions, while avoiding undesirable influence of these effects on editing-unrelated regions, thereby improving the editing effect and avoiding serious damage to the background. No matter which component disables the mask, it will cause the background preservation to be evidently worse, while the editing effect only has a marginal improvement, showcasing the effectiveness and necessity of the region-adaptive guidance.

To make the ablation studies clearer, we further provide qualitative visual comparisons of the editing results in Fig. C.3. The results in (a) indicate that, without the accurately reconstructable latent provided by Uni-Inv (utilizing a DDIM-Inversion-like inversion method), editing using flow models is likely to crash. Even though inversion using $\boldsymbol{v}(\cdot, t_i)$ has appropriately improved the inversion accuracy, it is still insufficient to support reliable real image editing. Meanwhile, the simple delayed injection without Uni-Edit provides low editing effects, resulting in unchanged images.

Furthermore, (b) shows the results after replacing the fused velocity $\boldsymbol{v}^F$ with $\boldsymbol{v}^T$ or $\boldsymbol{v}^S$. ① illustrates that directly utilizing $\boldsymbol{v}^T$ to move the sample can lead to the background not remaining unchanged. ② demonstrates that if we adopt only $\boldsymbol{v}^S$ as the velocity, even with the correction step, the results can be unnatural (it should have turned into a cup but did not). Additionally, the second row of (b) also indicates that velocity fusion can make the details of the results more reliable (velocity fusion provides the strawberry with a fuller color). Although the velocity itself may not have a strong impact on editing (just like the failure of delayed injection), velocity fusion can still enhance the editing details and provide regional sensitivity, thus making editing more precise and reliable.

Subsequently, (c) provides the editing results of using different inversion velocity functions. The velocity functions here are consistent with Fig. 4. It is evident that without precise inversion like Uni-Inv, achieving success in image editing is difficult, especially for flow models that are suscepti-

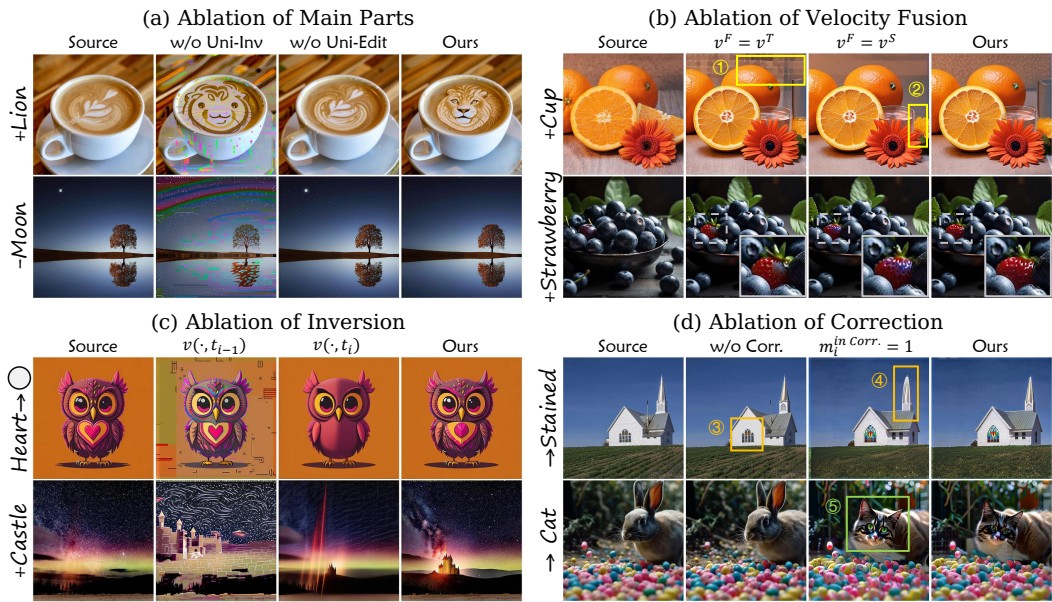

Figure C.3: **Qualitative comparison for ablation studies of Uni-Edit**. (a) Comparison of ablation in the main parts. (b) Comparison of different kinds of editing velocities. (c) Comparison of different kinds of inversion methods. (d) Ablation of correction in Uni-Edit.

ble to cumulative errors. Additionally, (d) presents the visualization of correction's ablation studies. "w/o Corr." means setting the correction step as $s_i = 0$, and $m_i^{\text{in Corr.}} = 1$ denotes turning the correction step into $s_i = \omega(t_{i-1} - t_i)(1+1) \odot v_i^-$. The former is similar to the simple delayed injection (instead of using fused velocity), which represents weakening the editing ability of Uni-Edit, while the latter means maintaining the editing ability of Uni-Edit but eliminating region-adaptive guidance. In these cases, ③ shows that without correction the method will significantly reduce editing ability, and ④ indicates that correction without region-adaptive guidance can easily cause changes in editing-irrelevant regions. Not only that, as shown in ⑤, correction without regional restrictions can also easily cause overexposure, which is similar to large classifier-free guidance (CFG). These visualizations of ablation results further demonstrate the roles and significance of the components in our proposed method.

### C.2 HYPER-PARAMETER SELECTION

Additionally, we present the ablation studies of our proposed Uni-Edit with step = 15 for various hyper-parameters in Fig. C.2. The results demonstrate that different hyper-parameters bring rational skews to the trade-off between background preservation and editing effectiveness. Nevertheless, our approach improves the overall level of the trade-off, making it effortless to bring benefits to both aspects.

## D UNI-INV ON DIFFERENT GENERATION METHODS

### D.1 HEUN METHOD BASED UNI-INV

To validate the transferability and effectiveness of our method on different samplers, we reimplement our Uni-Inv based on the Heun method. To be specific, since the Heun method is formulated as:

$$Z_{t_{i-1}} = Z_{t_i} + (t_{i-1} - t_i) \frac{v_\theta\left(Z_{t_i}, t_i\right) + v_\theta\left(v_\theta\left(Z_{t_i}, t_i\right), t_i\right)}{2}, \tag{D.18}$$

we directly reform the velocity function as:

$$v_\theta^H\left(\zeta, \tau\right) = \frac{v_\theta\left(\zeta, \tau\right) + v_\theta\left(v_\theta\left(\zeta, \tau\right), \tau\right)}{2}, \tag{D.19}$$

Table D.2: **Quantitative results for inversion and reconstruction** of our Uni-Inv based on Heun method with Flow models and DDIM with Diffusion models. We set the step to 50 for SDXL (Podell et al., 2023), 25 for SD3 (Esser et al., 2024), and 15 for FLUX models to conduct the experiments. The best results are **bolded**.

| Method | Model | Unconditional | | | | Conditional | | | |
|---|---|---|---|---|---|---|---|---|---|
| | | $MSE^{\downarrow}_{10^3}$ | $PSNR\uparrow$ | $SSIM^{\uparrow}_{10^2}$ | $LPIPS^{\downarrow}_{10^2}$ | $MSE^{\downarrow}_{10^3}$ | $PSNR\uparrow$ | $SSIM^{\uparrow}_{10^2}$ | $LPIPS^{\downarrow}_{10^2}$ |
| DDIM | SDXL | 8.99 | 22.19 | 75.57 | 12.76 | 7.35 | 23.21 | 77.73 | 10.20 |
| **Ours** (DDIM) | | **6.32** | **24.18** | **79.05** | **8.60** | **5.17** | **25.04** | **80.63** | **6.95** |
| Heun | SD3 | 25.34 | 16.98 | 67.25 | 26.63 | 26.32 | 16.89 | 64.14 | 27.70 |
| **Ours** (Heun) | | **20.23** | **20.10** | **76.38** | **15.76** | **12.75** | **22.31** | **79.62** | **12.43** |
| Heun | FLUX | 83.04 | 11.77 | 42.10 | 39.96 | 76.79 | 12.17 | 40.17 | 41.18 |
| **Ours** (Heun) | | **57.39** | **13.63** | **57.45** | **26.95** | **32.35** | **16.79** | **67.33** | **21.25** |

Table D.3: Inversion comparison between iterative inversion methods and Uni-Inv on the first 500 images of CC3M.

| Method | Model | Unconditional | | | Conditional | | | Steps | NFE |
|---|---|---|---|---|---|---|---|---|---|
| | | $PSNR\uparrow$ | $SSIM^{\uparrow}_{10^2}$ | $LPIPS^{\downarrow}_{10^2}$ | $PSNR\uparrow$ | $SSIM^{\uparrow}_{10^2}$ | $LPIPS^{\downarrow}_{10^2}$ | | |
| ReNoise | Diffusion | 24.14 | 78.71 | 11.99 | 24.29 | 78.95 | 11.66 | 50 | 150 |
| GNRI | Diffusion | 23.90 | 78.07 | 8.68 | 23.88 | 78.03 | 8.79 | 50 | 150 |
| **Ours** | Diffusion | **24.41** | **79.67** | **7.96** | **25.24** | **81.08** | **6.31** | 50 | 101 |
| ReNoise | SDXL-Turbo | 17.59 | 57.23 | 28.43 | 16.81 | 55.10 | 29.27 | 4 | 16 |
| GNRI | SDXL-Turbo | 13.46 | 49.43 | 39.82 | 13.20 | 48.56 | 38.68 | 4 | 12 |
| **Ours** | SDXL-Turbo | **20.08** | **71.15** | **14.32** | **20.63** | **71.86** | **13.14** | 4 | 9 |

then using $v_\theta^H$ to replace the original velocity function in Algo. 1. As shown in Tab. D.2, our approach improves the reconstruction accuracy of the Heun method across the board. This demonstrates the flexibility and adaptability of Uni-Inv to different samplers and reflects its effectiveness.

## D.2 DDIM BASED UNI-INV

DDIM (Song et al., 2020a) provides an efficient sampling method for the stochastic-differential-equation-based diffusion models and allows access to the sampling strategy with the form of ordinary differential equations. Benefiting from this, we are able to migrate Uni-Inv to diffusion models by simply treating the predicted initial noise as a velocity, utilizing the cached last-step predicted noise to push forward the current samples to the next timestep, thus performing our inversion. We evaluate the above strategy on SDXL (`RealVisXL_V4.0`) (Podell et al., 2023). The results are shown in Tab. D.2. Though DDIM has already demonstrated strong feasibility in numerous applications, our approach can still take it to the next level and bring about an overall improvement in reconstruction accuracy. It also indicates that our work does not just face a particular methodology. We expect to build approaches that can continuously provide insights into the developing trend of generative models.

## D.3 COMPARISON BETWEEN ITERATIVE INVERSION METHODS AND UNI-INV

We further compare iterative inversion methods (Garibi et al., 2024; Samuel et al., 2025) with our proposed Uni-Inv on inversion and reconstruction experiments. As there are no flow-based implementations of these methods, we compare inversion on diffusion using official code and settings in D.3. For fairness, we set the optimization steps on SDXL-Turbo of ReNoise (Garibi et al., 2024) to 2 as GNRI (Samuel et al., 2025) does. We adopt the same experimental settings as the manuscript, except for the sampling steps (50 for diffusion models and 4 for SDXL-Turbo). The experiments on

the first 500 samples of the CC3M (Sharma et al., 2018) dataset further provide Uni-Inv's superiority compared with the mentioned iterative inversion methods.

# E   Uni-Edit on Diffusion Models

Table E.4: **Text-driven image editing comparison** on PIE-Bench (Ju et al., 2024) based on Diffusion models. We evaluate our proposed Uni-Edit using SDXL (Podell et al., 2023). We keep the same hyper-parameter setting with our main experiments (*i.e.*, $\alpha = 0.6$ and $\omega = 5$), and adopt 50 and 15 as steps. Besides tuning-based methods are marked in gray, the best and second best results are **bolded** and underlined, respectively.

| Method | Model | Struc. Dist.$^{\downarrow}_{10^3}$ | BG Preservation | | | | CLIP Sim.$\uparrow$ | | Steps | NFE |
|---|---|---|---|---|---|---|---|---|---|---|
| | | | PSNR$\uparrow$ | LPIPS$^{\downarrow}_{10^3}$ | MSE$^{\downarrow}_{10^4}$ | SSIM$^{\uparrow}_{10^2}$ | Whole | Edited | | |
| Null-Text Inv | Diff. | 13.44 | 27.03 | 60.67 | 35.86 | 84.11 | 24.75 | 21.86 | 50 | - |
| ReNoise | Diff. | - | 27.11 | 49.25 | 31.23 | 72.30 | 23.98 | 21.26 | 50 | - |
| P2P | Diff. | 69.43 | 17.87 | 208.80 | 219.88 | 71.14 | 25.01 | 22.44 | 50 | 100 |
| P2P-Zero | Diff. | 61.68 | 20.44 | 172.22 | 144.12 | 74.67 | 22.80 | 20.54 | 50 | 100 |
| PnP | Diff. | 28.22 | 22.28 | 113.46 | 83.64 | 79.05 | 25.41 | 22.55 | 50 | 100 |
| PnP-Inv. | Diff. | 24.29 | 22.46 | 106.06 | 80.45 | 79.68 | 25.41 | 22.62 | 50 | 100 |
| EditFriendly | Diff. | - | 24.55 | 91.88 | 95.58 | 81.57 | 23.97 | 21.03 | 50 | 100 |
| MasaCtrl | Diff. | 28.38 | 22.17 | 106.62 | 86.97 | 79.67 | 23.96 | 21.16 | 50 | 100 |
| InfEdit | Diff. | **13.78** | **28.51** | **47.58** | **32.09** | **85.66** | 25.03 | 22.22 | 12 | 72 |
| **Ours** | Diff. | 15.59 | 25.64 | 78.83 | 43.05 | 83.42 | **26.33** | **22.78** | 15 | 28 |

Similar to Uni-Inv, since our proposed Uni-Edit is completely sample-based and model-agnostic, it is also capable of migrating to diffusion models effortlessly. We adopt the SDXL (RealVisXL_V4.0) (Podell et al., 2023) as our base model, conducting evaluation experiments on PIE-Bench (Ju et al., 2024). The results are shown in Tab. E.4. Here we enumerate the previous SOTA of diffusion-based approaches, wherein compared to the manuscript, we additionally present the results of tuning-based inversion (Mokady et al., 2023; Garibi et al., 2024) applied to editing. In contrast to these approaches, the CLIP similarity metrics exemplify our proposed Uni-Edit's capability to drive the diffusion-based editing to new heights and maintain highly competitive background preservation. Meanwhile, we significantly improve the editing efficiency by reducing NFE extremely compared to previous diffusion-based approaches. These experiments strongly demonstrate the effectiveness, adaptability, and generalizability of our proposed approaches, providing new insights into image inversion and editing in the era of flow models.

Furthermore, there are still many training-based methods (Wu et al., 2024; Brooks et al., 2023; Shi et al., 2024; Wei et al., 2024) to learn how to reasonably edit images from the provided training data. Most of them focus on conducting flexible editing through user-provided instructions. Such ideas are very practical and effective. Nonetheless, before embarking on this kind of approach, it is crucial to clearly learn about the properties of the base generation methods. This is also the main concentration of this paper.

# F   Diverse Applications

In addition to general text-driven image editing, the interpretable design of our method enables a wide range of applications. Fig. F.4 showcases its use for sketch-to-image (1st line) and stroke-to-image (2nd line) tasks (Yu et al., 2015; Chowdhury et al., 2022). For these applications, we set $\alpha = 0.8$ to enhance the editing effect. By exploiting the binary nature of sketches and fixing $m_i$ as their grayscale value, we achieve more robust results for sketch-to-image tasks. Moreover, thanks to the advanced flow matching-based video generation model Wan (WanTeam et al., 2025), we further test Uni-Edit on video editing tasks. We directly consider the latent containing the temporal dimension as $Z$, and apply Uni-Edit to Wan's sampling process without any modification. Setting $\alpha = 0.8$ and

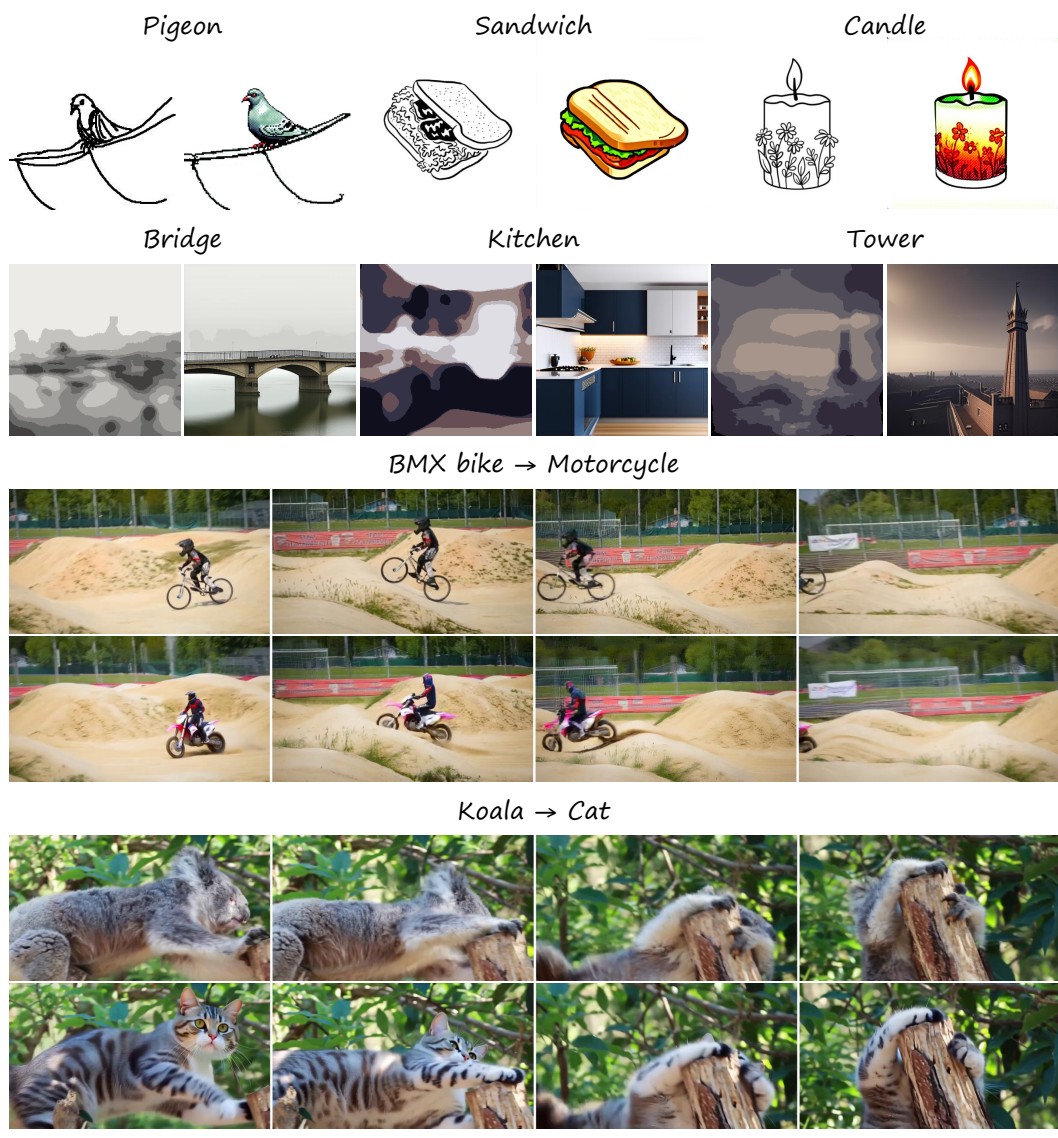

Figure F.4: **Diverse application of Uni-Edit**. The top is sketch to image, and the bottom is stroke to image. The left of a image pair is the source image, and the right is the editing result.

$N = 25$, we achieve reliable editing results (3rd and 4th parts) using `Wan2.1-T2V-1.3B` model. These results further highlight the generalizability and effectiveness of our approach.

## G APPLICATION UTILIZING DIVERSIFIED PLUGINS

Since our proposed method is model-agnostic, various plugins that can insert flow models can be applied to provide different editing conditions or to achieve specific editing objectives. These plugins can generally provide *new conditions* or help enhance *controllability*, enabling image editing to meet different specific needs. Therefore, in the era of rapid development of generative models represented by flow models, our method can stably and continuously integrate into stronger models or more complex tasks.

### G.1 INTRODUCING OF NEW CONDITIONS

Many previous works achieve personalized generation based on images by inserting image features into prompt embeddings or attentions, among which IP-Adapter (Ye et al., 2023) is one of the most

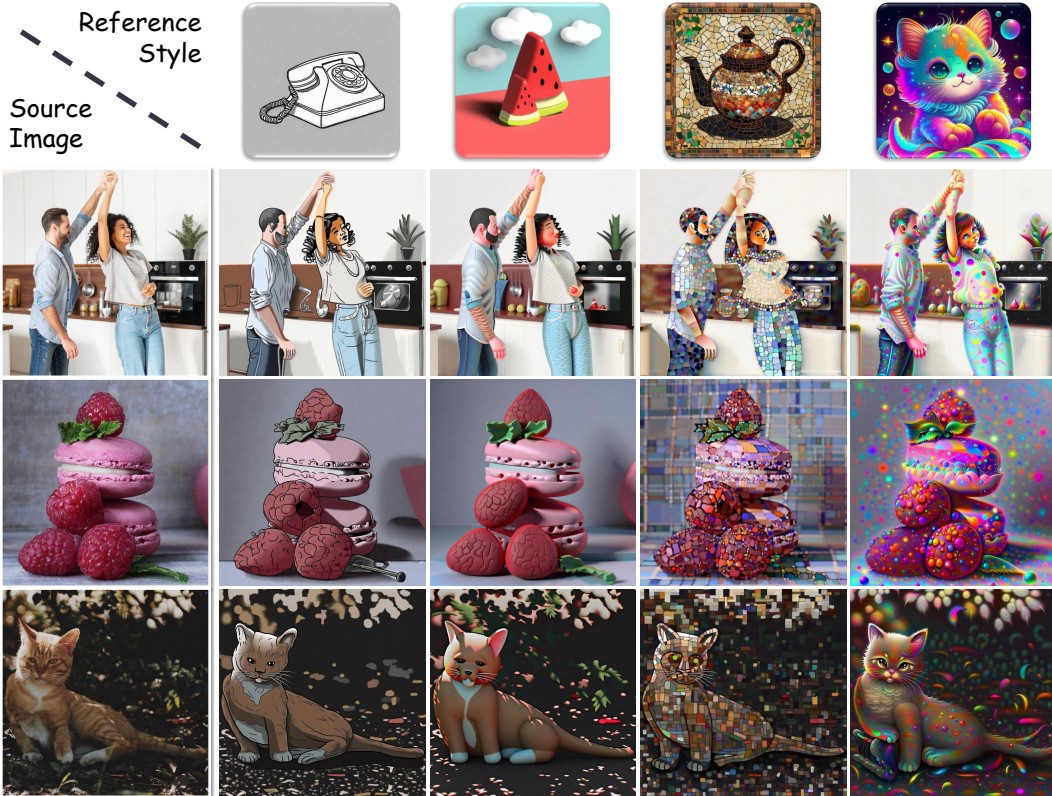

Figure G.5: **Applications of Uni-Edit utilizing IP-Adapter (Ye et al., 2023; Team, 2024) for reference-based style transfer**. The first column is the source image, and the first row is the reference style image.

representative approaches. Taking inspiration from this, we first attempted to apply an IP-Adapter that facilitates style transformation (Team, 2024) to our pipeline. During the editing process of Uni-Edit, we load `InstantX/FLUX.1-dev-IP-Adapter` into the FLUX model and differentiate between source and target conditions by distinguishing among different image inputs.

Specifically, we first employ the original Uni-Inv conditioned on null text, and then modify the $v_i^S$ and $v_i^T$ in Uni-Edit. After adopting the IP-Adapter, the velocity function becomes $v = v_\theta(Z_t, t \mid c_{\text{txt}}, c_{\text{img}})$, where $c_{\text{img}}$ denote the input image of the IP-Adapter. Subsequently, in order to make the image editing focus on style transfer, we keep the text conditions of $v_i^S$ and $v_i^T$ consistent with $c_{\text{txt}}^S$, without introducing any changes to the content:

$$v_i^S = v_\theta(\widetilde{Z}_{t_i}, t_i \mid c_{\text{txt}}^S, c_{\text{img}}^S), \quad v_i^T = v_\theta(\widetilde{Z}_{t_i}, t_i \mid c_{\text{txt}}^S, c_{\text{img}}^T). \quad (G.20)$$

Fig. G.5 shows the results of style transfer using our proposed Uni-Edit on IP-Adapter-injected FLUX model. We set $\alpha = 0.6, \omega = 5.0, N = 15$ for Uni-Edit here. We adopt the source image as $c_{\text{img}}^S$ and the reference style image as $c_{\text{img}}^T$. It can be clearly observed that the editing results accurately capture the style of the reference style image (such as 3D rendering style, colorful style, etc.). At the same time, our method does not cause excessive damage to the source image, allowing the edited results to maintain both the targeted style features and the original content.

In addition, under the same paradigm, we have also adopted another type of IP-Adapter, InstantCharacter (Tao et al., 2025), which has the ability to customize the characters in the generated images. We utilize the source image as $c_{\text{img}}^S$ and the reference character image as $c_{\text{img}}^T$ to achieve image character editing, and set $\alpha = 0.7, \omega = 3.0, N = 30$ here. The results are shown in Fig. G.6, which showcases the abilities of our method using InstantCharacter for effective face editing.

These experiments not only demonstrate the strong flexibility and diverse application scenarios of our proposed method, but also illustrate the promising scalability of such a sampling based strategy.

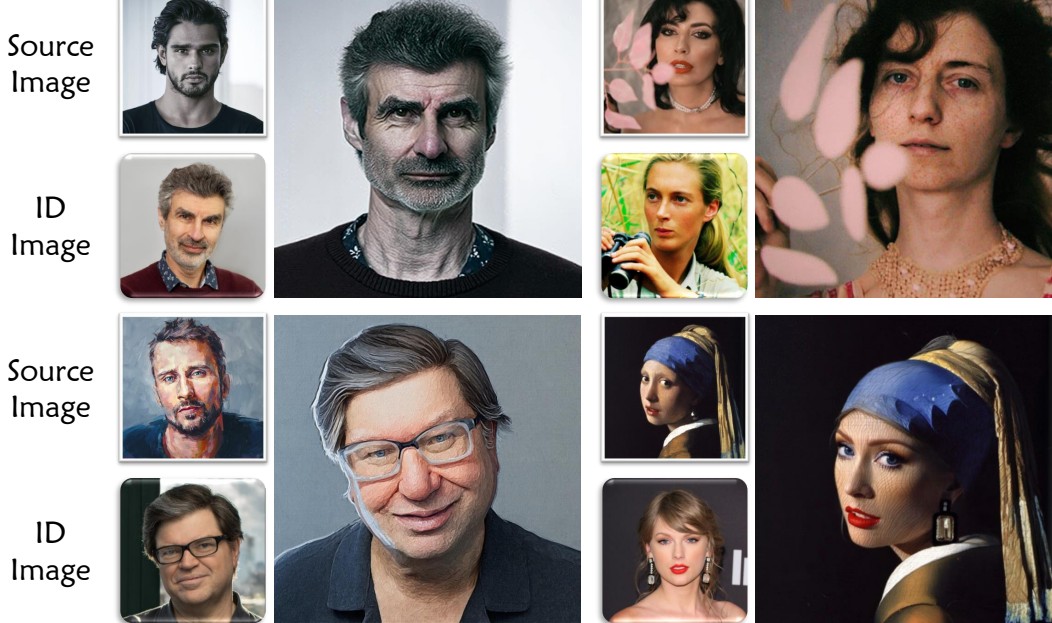

Figure G.6: **Applications of Uni-Edit utilizing InstantCharacter (Tao et al., 2025) for face editing**. In each group, the upper left is the source image, the lower left is the reference character image, and the right is the editing result.

## G.2 ENHANCEMENT OF CONTROLLABILITY

Previous work has also proposed many modules that inject additional control conditions, among which ControlNet (Zhang et al., 2023) is the most representative. We can introduce effective control during the editing process of Uni-Edit by treating these injected conditions as part of the velocity function, *i.e.*:

$$\boldsymbol{v}_\theta'(\widetilde{\boldsymbol{Z}}_t, t \mid \boldsymbol{c}_{\text{txt}}) = \boldsymbol{v}_\theta(\widetilde{\boldsymbol{Z}}_t, t \mid \boldsymbol{c}_{\text{txt}}, \boldsymbol{c}_{\text{ctrl}}), \tag{G.21}$$

where $\boldsymbol{c}_{\text{ctrl}}$ denotes the input condition of the ControlNet. By replacing $\boldsymbol{v}_\theta$ in Alg. 2 with $\boldsymbol{v}_\theta'$, it can be ensured that the control conditions are preserved in the image during the inversion and editing process.

Fig. G.7 shows the editing results with enhanced controllability. We utilize Stable Diffusion 3 with Canny-conditioned ControlNet (`InstantX/SD3-Controlnet-Canny`) as the base model for our Uni-Inv and Uni-Edit, and set $\alpha = 0.9, \omega = 5.0, N = 30$. These images are from the GTAV dataset (Richter et al., 2016), and the Canny edges used for control are the Canny edges of the segmentation labels of these images. We utilize null text as the source prompts and the word describing environment ("Snowy", "Rainy", "Foggy", and "Night") as the target prompts in these experiments. The results indicate that after introducing the control of ControlNet, Uni-Edit exhibits strong semantic information retention abilities and also achieves significant editing of environmental features in the image. This application can be used in autonomous driving scenarios or world model building processes to provide more diverse while reliable data for training semantic segmentation and detection recognition models.

## H ADDITIONAL QUALITATIVE COMPARISON

### H.1 UNI-INV

We represent more qualitative comparison results of our Uni-Inv and recent flow-based approaches (Wang et al., 2024b; Deng et al., 2024) in Fig. H.8 and Fig. H.9. These figures contain a wide variety of image samples, including landscape photographs, object photographs, human-centered daily photographs, photographs in extreme lighting, group photographs of large numbers of people,

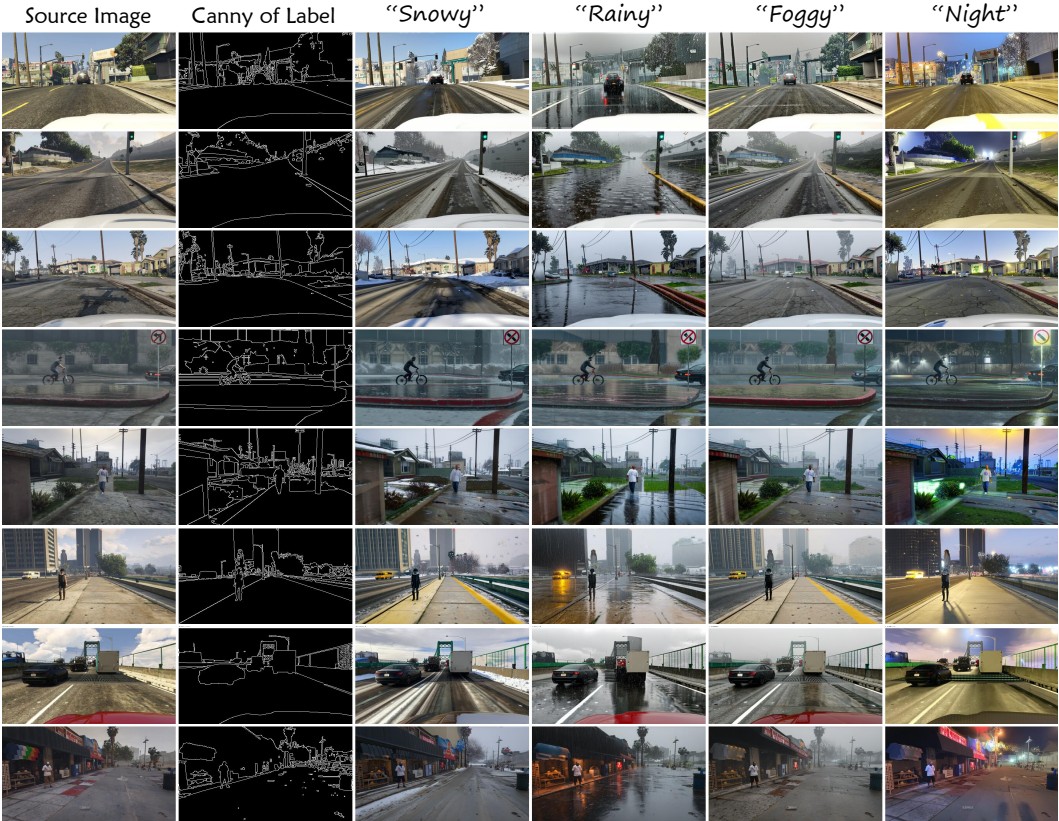

Figure G.7: **Application of Uni-Edit for reliable environment style transformation for autonomous driving tasks using ControlNet (Zhang et al., 2023).** The first column is the source image, and the second column is the reference canny image, which is obtained from the ground truth segmentation label of the source image. The first row provides editing prompts of such editing tasks (only one discription word is enough).

black and white photographs, posters, pencil drawings, oil paintings, etc. of varying resolutions. Our method well maintains the overall image color (last line of Fig. H.8), texture style (6th line of Fig. H.8), content details including text (8th and 9th lines of Fig. H.9) during inversion & reconstruction, achieving consistent superiority in both conditional and unconditional settings.

## H.2 UNI-EDIT

We further perform additional qualitative comparisons with existing state-of-the-art methods (Hertz et al., 2022; Tumanyan et al., 2023; Ju et al., 2024; Huberman-Spiegelglas et al., 2024; Cao et al., 2023; Xu et al., 2024a; Rout et al., 2024; Wang et al., 2024b; Deng et al., 2024) on text-driven image editing as shown in Fig. H.10, Fig. H.11, and Fig. H.12. We extensively compared the different approaches under conditions of editing categories, materials, properties, motions, backgrounds, and types of adding or removing items or concepts, as well as stylization.

First, in the task of regional editing, our method demonstrates significant local perception and background preservation capabilities. It is worth noting that our approach is model-agnostic, *i.e.* it does not require the involvement of the attention mechanism. This leads to more oriented regional editing. For example, attention-based methods such as PnP (Tumanyan et al., 2023) often impose attributes that need to be used for regional editing on irrelevant regions (4th line of Fig. H.11). Due to these methods disassembling prompts and attempting to utilize a single token about the editing to exert guidance, inappropriate semantic understanding comes since the wholeness of prompts is destroyed. On the contrary, our approach is model-agnostic, thus better capitalizing on the text comprehension capabilities learned from the model.

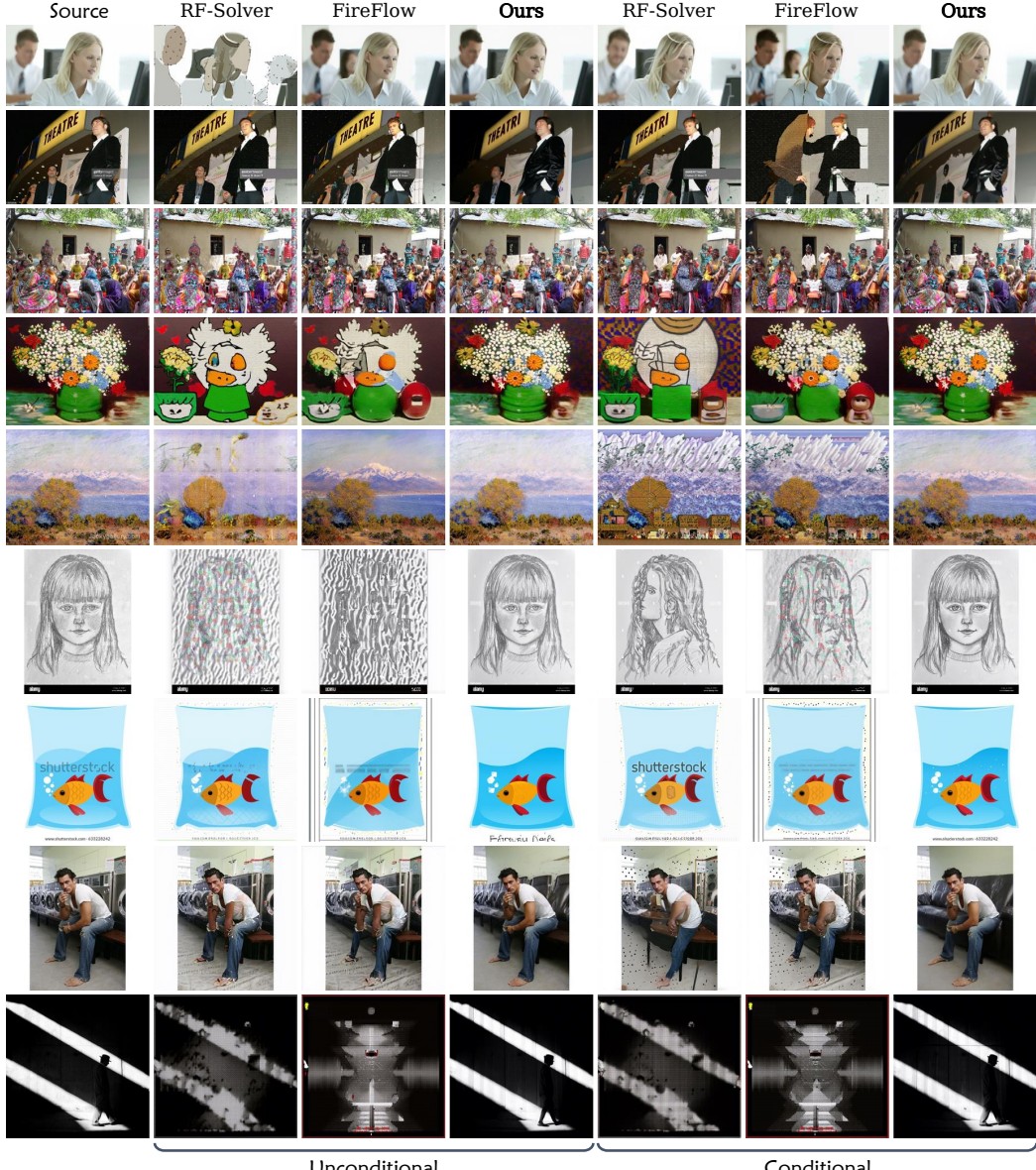

Figure H.8: **Additional qualitative comparison on inversion & reconstruction** on the Conceptual Captions validation dataset (Sharma et al., 2018).

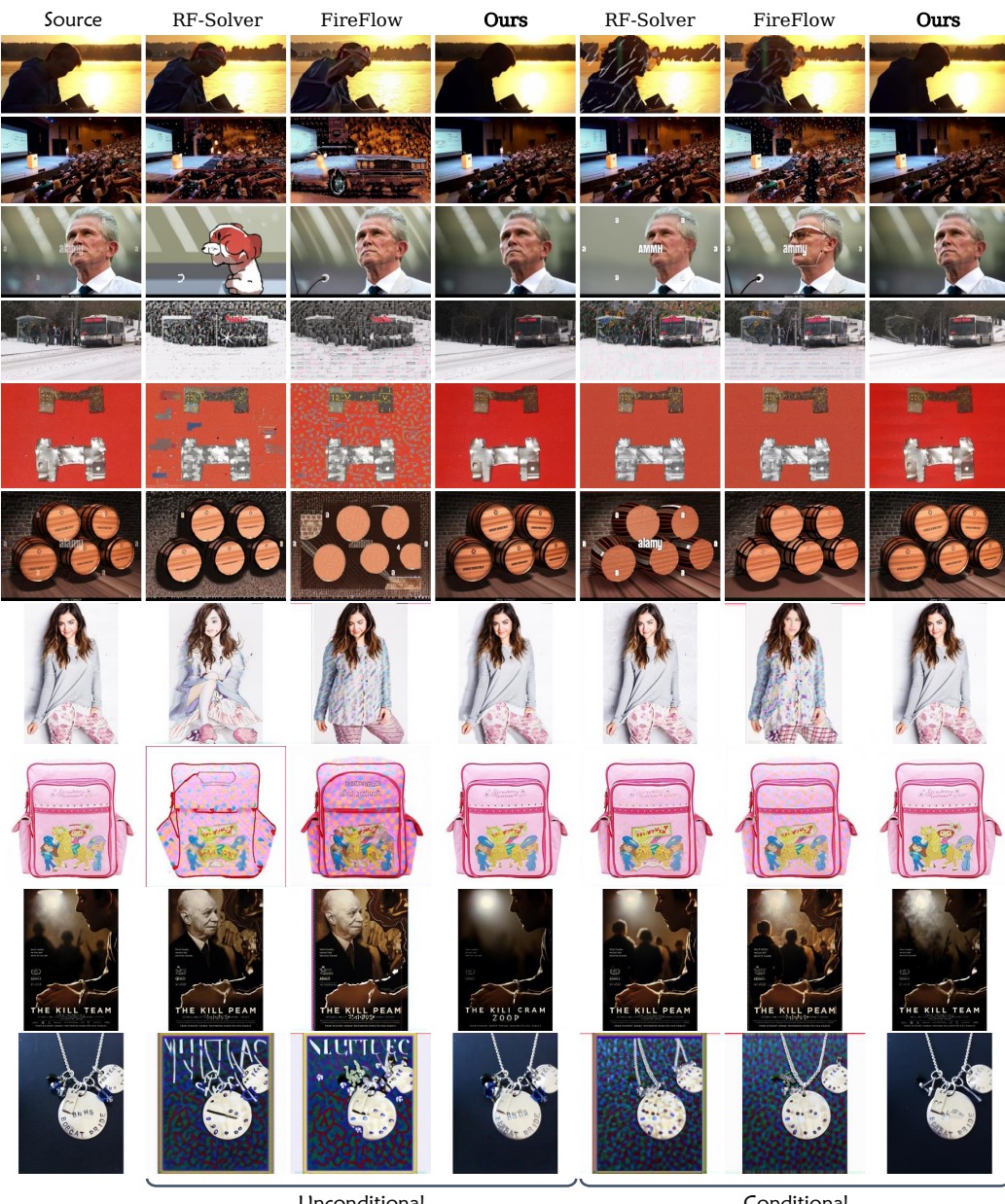

Figure H.9: **Additional qualitative comparison on inversion & reconstruction** on the Conceptual Captions validation dataset (Sharma et al., 2018).

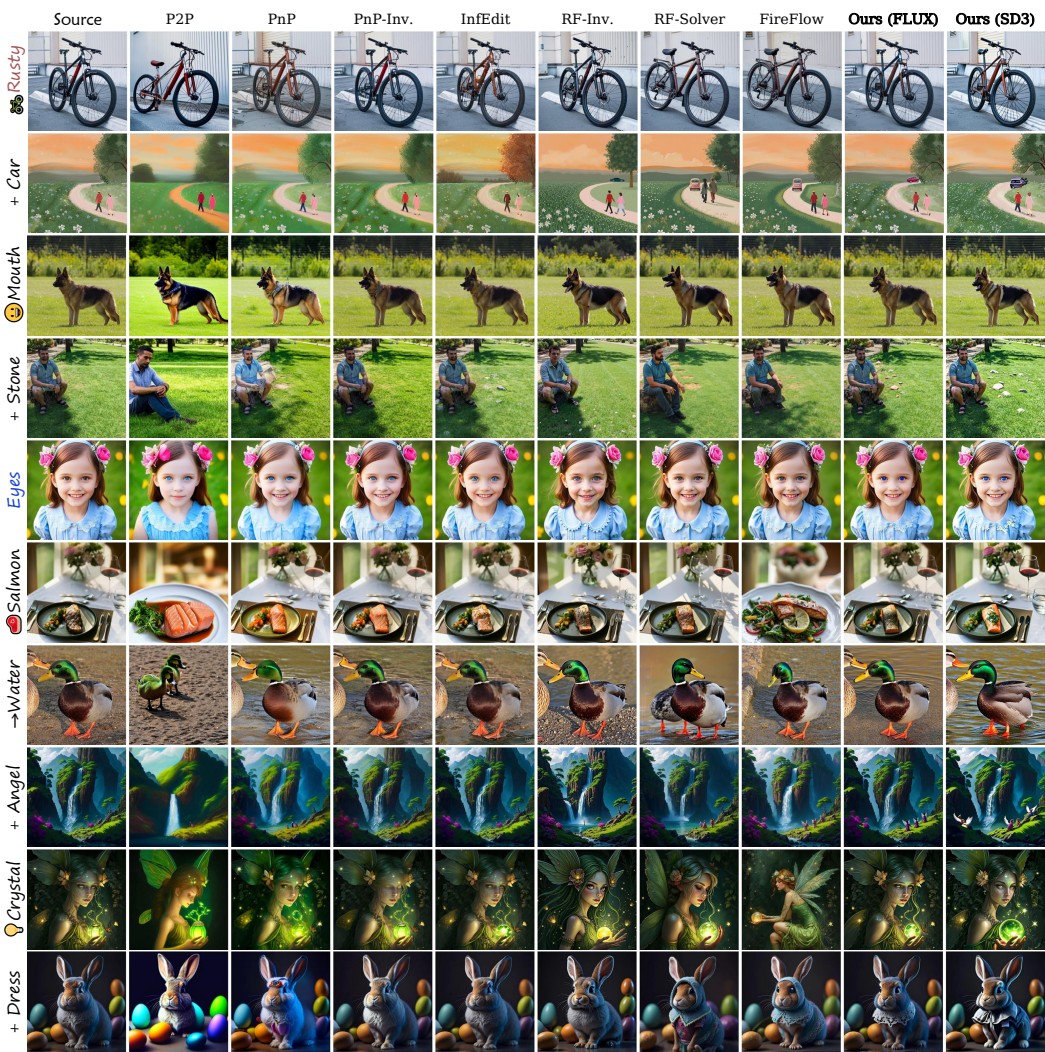

Figure H.10: **Additional qualitative comparison on image editing** on PIE-Bench (Ju et al., 2024).

Subsequently, compared to the same sampling-based kind of approaches (*i.e.*, RF-Inversion (Rout et al., 2024), RF-Solver (Wang et al., 2024b), and FireFlow (Deng et al., 2024)), our method demonstrates significant advantages in terms of image structure and background preservation while maintaining robust editing (1st and 9th lines of Fig. H.10, 3rd and last lines of Fig. H.11, 3rd and 7th lines of Fig. H.12). On the one hand, this is due to our proposed Uni-Inv theoretically ensuring a small local error in the inversion process that can support accurate reconstruction. On the other hand, our deep exploration and re-empowerment of delayed injection make it easy for our proposed Uni-Edit to strike a satisfying balance between editing and the preservation of editing-irrelevant concepts.

# I ADDITIONAL RESULTS ON EDITING TASKS

## I.1 IMAGE EDITING

Fig. I.13 and Fig. I.14 represent additional qualitative results of our proposed Uni-Edit on image editing tasks. These results indicate that when it comes to diverse targets and diverse image domains, our approach still remains very effective. It is worth noting that each edit in Fig. I.13 contains multiple different editing objectives (*e.g.*, changing the time, removing the crowd, and adjusting the lighting). Our approach is able to simultaneously achieve these various targets in a single round, using only the original prompt and the target prompt as guidance. Benefiting from the sampling-

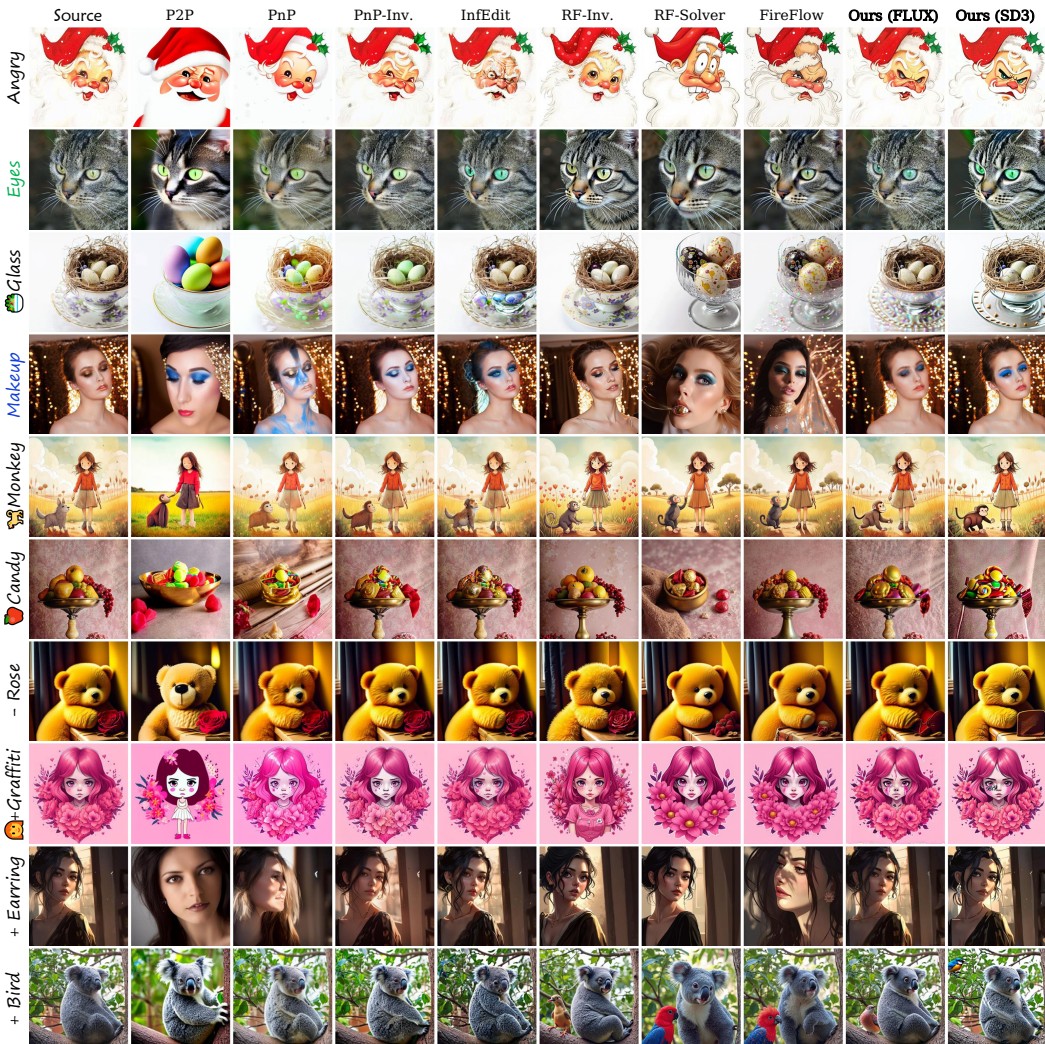

Figure H.11: **Additional qualitative comparison on image editing** on PIE-Bench (Ju et al., 2024).

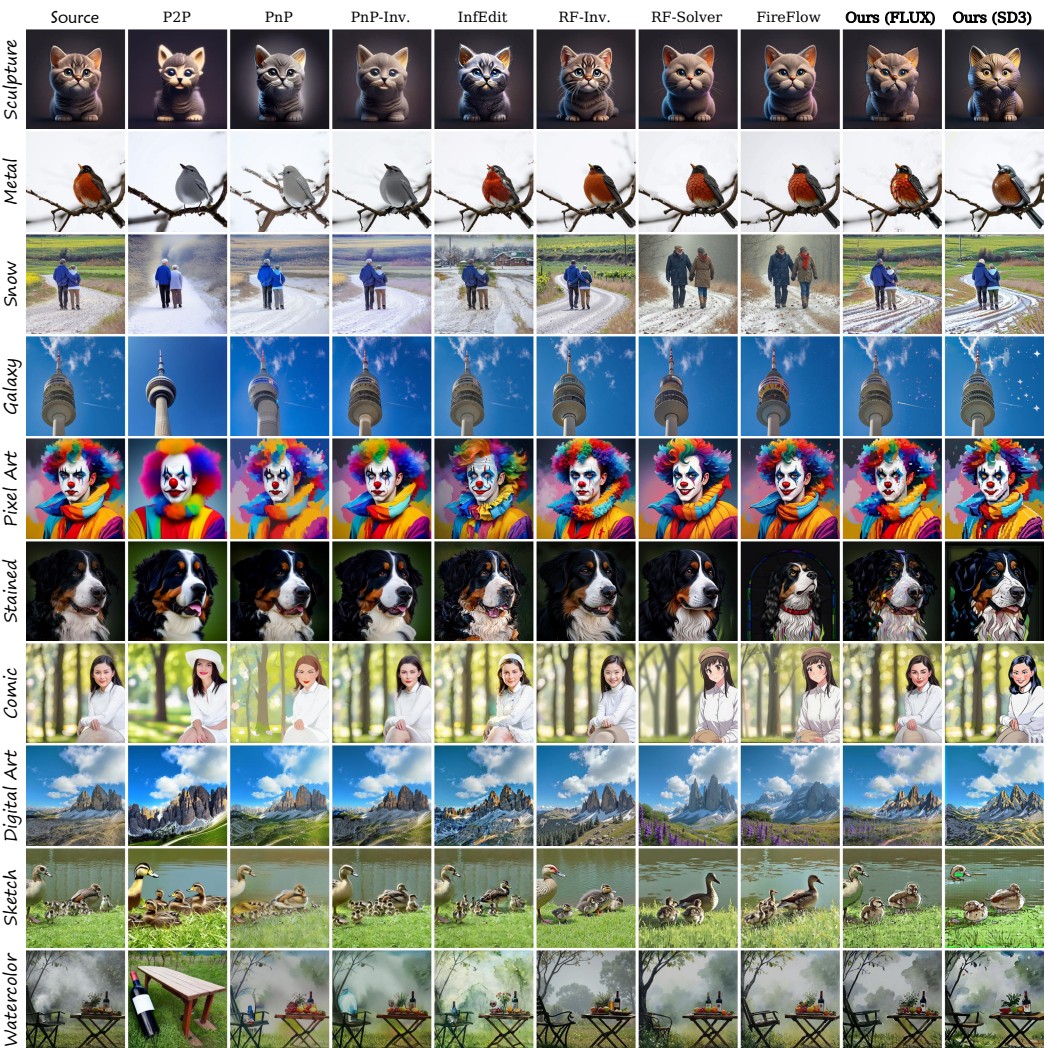

Figure H.12: **Additional qualitative comparison on image editing** on PIE-Bench (Ju et al., 2024).

based design, our approach is able to capture diverse objectives at once through the relationship between the latents obtained from different conditions. Compared to methods that rely on cross attention manipulations, this design is simpler, more robust, and less likely to cause confusion.

### I.2 VIDEO EDITING

Moreover, we directly adopt Uni-Edit to conduct video editing tasks using the flow matching-based video generation model Wan (WanTeam et al., 2025). Qualitative results are shown in Fig. I.15. Since our method is model-agnostic, we can achieve reliable video editing results without additional design or complex parameterization. It is further strong evidence of our approach's generalizability.

## J LIMITATIONS AND FUTURE WORKS

The core issue plaguing us now is that our Uni-Edit is designed for image-text pair inputs. It is not capable of accepting more than one image as the condition. This results in no direct way for us to contribute to the personalization generation problems. In the future, we would like to develop editing methods that are more general and oriented to more diverse tasks. The accurate inversion of Uni-Inv helps to capture image information. With this facilitation, we hope to develop sampling-based editing strategies capable of injecting image conditions based on our re-enabled delayed injection framework. We leave it as an interesting future work.

## K LLM USAGE STATEMENT

In this paper, LLMs were not used for polishing writing, discovery and retrieval, research ideation, and other aspects. All paper writing, scientific content, and interpretations are the authors' own.

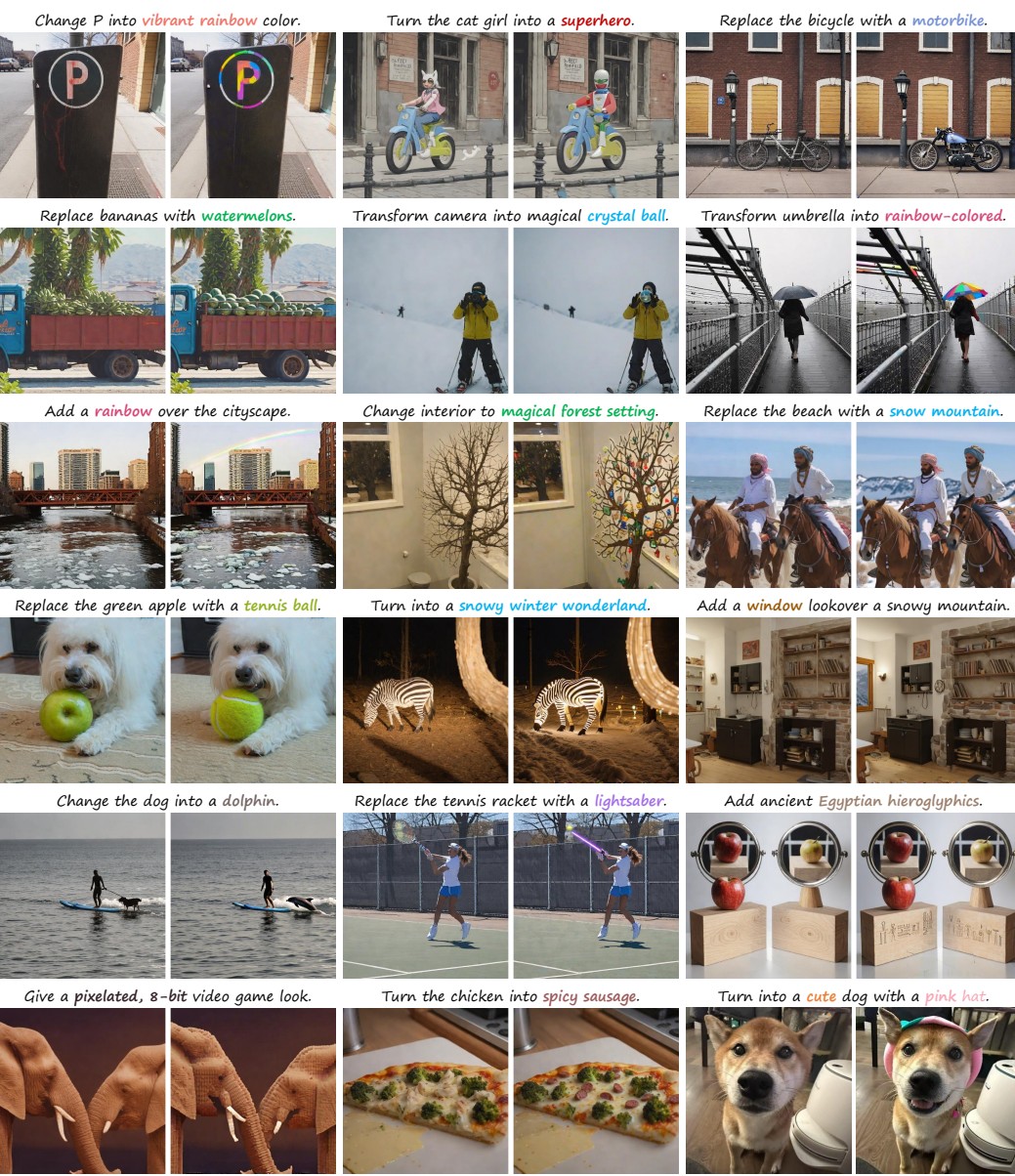

Figure I.13: **Additional qualitative results on image editing** on UltraEdit dataset (Zhao et al., 2024) and wild images. In each image pair, the left is the original image and the right is the result of our editing. The text captions at the top of images are descriptions of the editing objectives and are not the input to the model. We still maintain the paradigm of using the original prompt and the target prompt as conditions. These images are obtained by FLUX using Uni-Edit with $\alpha = 0.6$, $\omega = 5$ and step $= 15$ which is consistent with the main experiments.

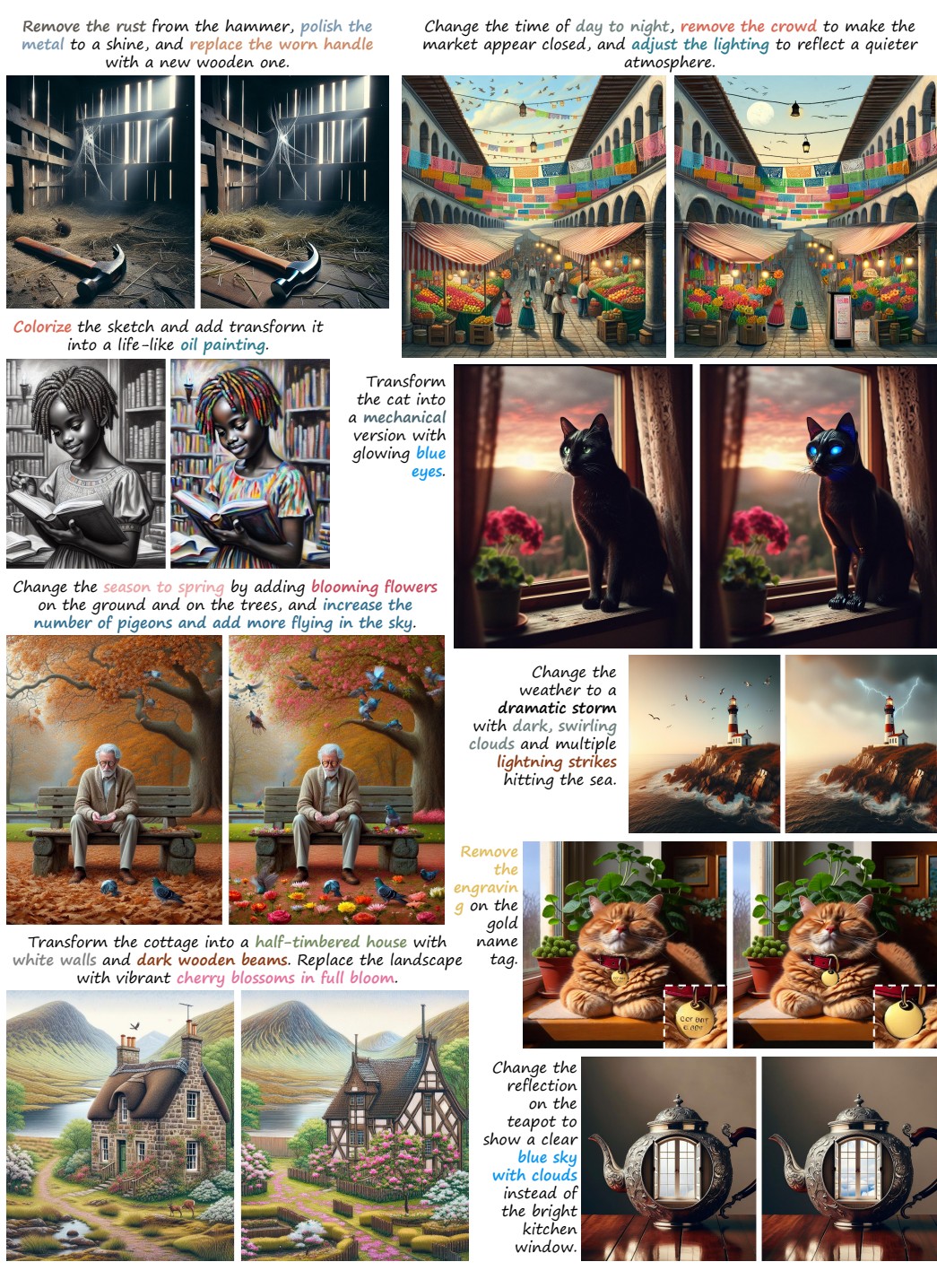

Figure I.14: **Additional qualitative results on image editing** on HQ-Edit dataset (Hui et al., 2024). The visualization setup is the same as Fig. I.13.

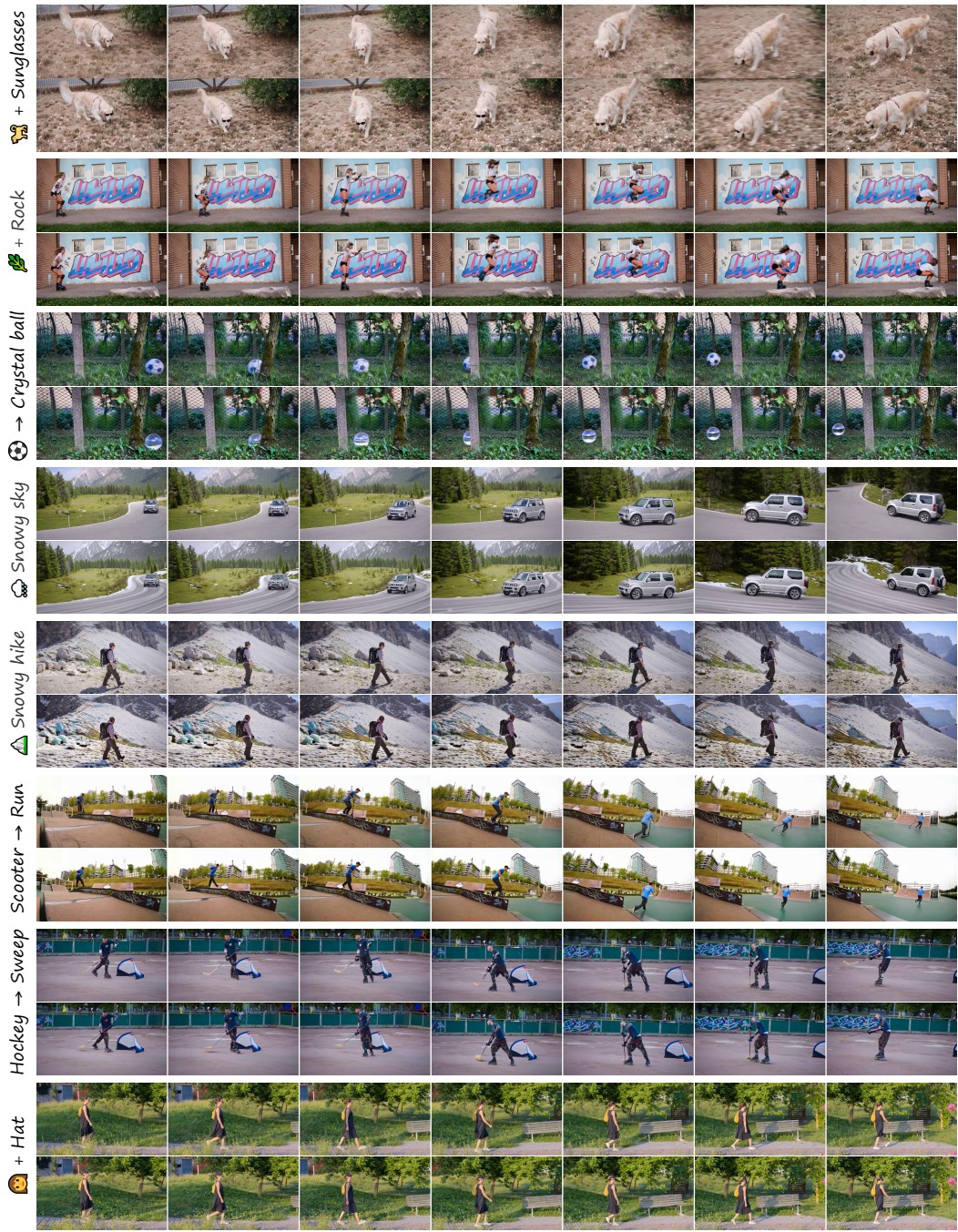

Figure I.15: **Additional qualitative results on video editing** on DAVIS dataset (Pont-Tuset et al., 2017). We set $\alpha = 0.8$, $\omega = 5$, $N = 25$ for the experiments.

