# OpenReview forum: "UniEdit-Flow: Unleashing Inversion and Editing in the Era of Flow Models"
_ICLR.cc/2026/Conference — ICLR 2026 Poster_

### Official Review · Reviewer_7YEx · 2025-10-30

**Soundness:** 3
**Presentation:** 3
**Contribution:** 2
**Rating:** 6
**Confidence:** 3

**Summary:**

The paper proposes Uni-Inv and Uni-Edit. Uni-Inv is a predictor-corrector based inversion method that achieves accurate image reconstruction from latent noise, mitigating the large reconstruction errors seen in vanilla flow inversion. Building on this, Uni-Edit is a region-aware image editing strategy that re-enables the concept of delayed injection for flow models. Both methods obtain SoTA results on inversion and editing respectively.

**Strengths:**

1. The logic is reasonable and the writing is good.
2. Uni-inv and uni-edit applies the design of "take a step back" and "take a step forward", which has emperically demonstrated strong performance.
3.  The paper provides a theoretical analysis of Uni-Inv, bounding its local error to $\mathcal{O}(\Delta t_{i}^{3})$, which justifies its high reconstruction quality.

**Weaknesses:**

1. The reliability of the method seems to be dependent on the choice of hyper-parameter, different images might have different choices of hyper-parameter to achieven the best results. I'm pointing it out since this is a general problem for inversion based editing methods.
2. On the application side, could Uni-Inv's accurate latent noise capture be used in conjunction with a framework like IP-Adapter, applied to the flow latent, to enable this image-conditioned editing?
3. The region-adaptive mask m_{i} is calculated based on the difference between target and source velocities. Did the authors experiment with incorporating attention maps, as commonly done in diffusion editing, to see if an even more semantically precise mask could be generated?

**Questions:**

see weakness section.

---

> ### Author Response · Authors · 2025-11-21
> **Response to Reviewer 7YEx**
>
> Dear Reviewer 7YEx,
>
> Thanks for your appreciation of our paper writing, method design, and theoretical analysis. The questions you raised about hyper-parameter and further application of the proposed method are very insightful. We now respond to the individual questions below.
>
> ---
>
> **[W1] The reliability of the method seems to be dependent on the choice of hyper-parameter, different images might have different choices of hyper-parameter to achieven the best results. I'm pointing it out since this is a general problem for inversion based editing methods.**
>
> Thank you for your comment. You are right, and whether it is an inversion-based or inversion-free editing method, hyper-parameter sensitivity is always a common issue. In previous papers such as LEDITS++ [3], the various methods compared seemed to be based on hype-parameter search and selecting the best reconstruction or editing results for each sample separately.
>
> But in our manuscript, we have **fixed the hyper-parameters across different images** in the entire benchmark evaluation, without any sample-wise hyper-parameter selection. For different methods including our proposed method and baseline approaches, we still do some benchmark-wise hyper-parameter search or adopt the best hyper-parameters reported in their papers to **ensure the comparison is fair**.
>
> [3] LEDITS++: Limitless Image Editing using Text-to-Image Models.
>
> ---
>
> **[W2] On the application side, could Uni-Inv's accurate latent noise capture be used in conjunction with a framework like IP-Adapter, applied to the flow latent, to enable this image-conditioned editing?**
>
> Thank you for your insightful advice. **The application you mentioned is certainly possible**. Just like how we use Wan for video editing in our application, as long as the model itself is an iterative generative model, our method can be seamlessly integrated. Just as our motivation, we hope to propose a set of inversion and editing methods that can continuously expand their functionalities in the era of rapidly developing iterative generative models.
>
> In order to validate your suggestions, we have applied multiple plugins for diverse applications and shown their visual results in **Sec. G** in appendix:
>
> - **Fig. G.5**: Applications of Uni-Edit utilizing IP-Adapter [4] for **reference-image-based style transfer**.
> - **Fig. G.6**: Applications of Uni-Edit utilizing InstantCharacter [5] for **image-conditioned face editing**.
> - **Fig. G.7**: Application of Uni-Edit for **reliable environment style transformation** using ControlNet [6] for autonomous driving tasks.
>
> These results demonstrate that our proposed method has strong flexibility and generalization capability, which allows it to gradually integrate into more interesting applications as the iterative generation model develops.
>
> [4] IP-Adapter: Text Compatible Image Prompt Adapter for Text-to-Image Diffusion Models.
> [5] InstantCharacter: Personalize Any Characters with a Scalable Diffusion Transformer Framework.
> [6] Adding conditional control to text-to-image diffusion models.
>
> ---
>
> **[W3] The region-adaptive mask m\_{i} is calculated based on the difference between target and source velocities. Did the authors experiment with incorporating attention maps, as commonly done in diffusion editing, to see if an even more semantically precise mask could be generated?**
>
> Thank you for your suggestions and outlook. Many existing works have shown that reliable masks can be obtained from attention maps. If a more accurate mask can be obtained, editing will undoubtedly achieve better results.
>
> In this paper, we did not adopt attention-related operations as we would like our method to be robust to the **architectural change** brought by the evolution of flow models. We believe that the model architectural itself will continue to evolve, and many old methods are easily outdated as a result. We hope that the method we are building is agnostic to a particular type of architecture, but applicable to most of the mainstream generative models. Meanwhile, we also believe that the attention operation you mentioned, like the module you suggested in [W2], can be naturally integrated into our method as an expansion service under specific targets.
>
> ---

---

> > ### Comment · Reviewer_7YEx · 2025-11-27
> >
> > Thanks the authors for answering the questions and addressing my concerns. I will keep my positive rating towards acceptance.

---

> > > ### Author Response · Authors · 2025-11-28
> > > **Response to Reviewer 7YEx's feedback**
> > >
> > > Thank you for your insights that have helped us largely diversify our proposed method's applications. Thank you again for your recognition.

---

### Official Review · Reviewer_xe1o · 2025-10-31

**Soundness:** 3
**Presentation:** 3
**Contribution:** 2
**Rating:** 4
**Confidence:** 4

**Summary:**

The paper proposes a new method for image inversion and then editing for flow based models. Compared with previous DDIM inversion style 1st order inversion, the proposed method can be as a high order method. Results show improvement on image editing benchmark PIE-bench.

**Strengths:**

1. In general, the paper is well-written and easy to follow.
2. The results on PIE-Bench are quite good compared with previous methods.

**Weaknesses:**

1. Overall the proposed inversion method seems a special type of Heun. The difference is: Heun uses the average slope (algo 1 line 3 in the "for" loop) while the proposed method only uses the corrector slope.

2. Continue from 1, therefore, I do not fully understand why the proposed method can do better than Heun. Based on Prop 4.1 the local error is of O(t^3). Heun should have the same bound. But based on the visualizations and results from the tables, Heun is not as good as the proposed method.

3. Can you explain appendix section D. Why the inversion method should be related to samplers? The purpose of inversion and inversion based editing is to find the noise given an existing real world image and we are not suppose to know how the image is generated.

4. Continue from 3, what is the data you used in Table 1? Are they generated or real world data? Are they generated with a certain type of solver?

5. Some of the claims are questionable. For example, the trajectory of flow models is not straight, only the conditional trajectory is straight.

6. The authors may consider to discuss some related to works for inversion in flow based models. For example [1][2]

[1] inversion free image editing (ICCV 25)
[2] text-to-image rectified flow as plug-and-play priors (ICLR 25)

**Questions:**

I include the questions in the weaknesses part. Currently I feel the performance is good but I do not fully understand the difference between the proposed method and other high-order methods (or Is it just a reuse of heun-like method for inversion?).

I'll consider to increase my score if my concerns are addressed in the rebuttal.

**Details Of Ethics Concerns:**

N.A.

---

> ### Author Response · Authors · 2025-11-21
> **Response to Reviewer xe1o (1)**
>
> Dear Reviewer xe1o,
>
> Thanks for your constructive feedback on our work. Your questions and suggestions help us explain the methods better and more intuitively. We respond to your individual questions below.
>
> ---
>
> > **[W1] Overall the proposed inversion method seems a special type of Heun. The difference is: Heun uses the average slope (algo 1 line 3 in the "for" loop) while the proposed method only uses the corrector slope.**
> >
>
> Thanks for your comment. However, **Uni-Inv is a separate inversion methodology rather than a solver**. Specifically, if follow the special type of Heun (referred to as $\text{Heun}^{\S}$) you mentioned, the corresponding steps in the inversion and sampling are:
>
> $$
> \text{{Heun}}^{\S}:~ Z\_{t\_{i-1}}=Z\_{t\_i}+(t\_{i-1}-t\_i)\underline{v\_{i-1}}, \\
> \text{Inv.}:~ \widehat{Z}\_{t\_i}=\widehat{Z}\_{t\_{i-1}} - (t\_{i-1}-t\_i)\widehat{v}\_i,
> $$
>
> where $\underline{v\_{i-1}}$ also requires second-order methods to obtain in the denoising process. But **Uni-Inv provides a inversion process to obtain inverted noise for a given sampler by "take a step back then take a step forward”**. Take the Euler method as an example, the corresponding steps are:
>
> $$
> \text{Euler}:~ Z\_{t\_{i-1}}=Z\_{t\_i}+(t\_{i-1}-t\_i)\underline{v\_{i}}, \\
> \text{Uni-Inv}:~ \widehat{Z}\_{t\_i}=\widehat{Z}\_{t\_{i-1}} - (t\_{i-1}-t\_i)\widehat{v}\_i.
> $$
>
> The difference is marked with the underline, showing that the sampling process is still first-order. In a nutshell, Uni-Inv is not used as a solver, but rather as an **inversion methodology for accurate reconstruction**, aimed at providing accurate inversion noise to a specified sampler (such as Euler, Heun).
>
> ---
>
> > **[W2] Continue from 1, therefore, I do not fully understand why the proposed method can do better than Heun. Based on Prop 4.1 the local error is of O(t^3). Heun should have the same bound. But based on the visualizations and results from the tables, Heun is not as good as the proposed method.**
> >
>
> Thanks for your in-depth comment. Solvers focuses on how to discretize a continuous differential equation to make the solving process from noise $Z\_1$ to data $Z\_0$ more accurate. Therefore, the error of the high-order solver you mentioned refers to the **discretization error**:
>
> $$
> \mathcal{E}^{dis}=\Vert \text{Solver}^{dis}(Z\_1) - \text{Solver}^{con}(Z\_1) \Vert,
> $$
>
> where $\text{Solver}^{dis}$ and $\text{Solver}^{con}$ means the discretized and the continuous solver, respectively.
>
> Unlike above, the error in our analyses is the **reconstruction error**. Specifically, what Uni-Inv want to do is that, given $Z\_0$ and a sampler, we want to obtain an inversion process to minimize the reconstruction error:
>
> $$
> \mathcal{E}^{rec}=\Vert Z\_0 - \text{Sampler}(Z\_1) \Vert = \Vert Z\_0 - \text{Sampler}(\text{Inversion}(Z\_0)) \Vert.
> $$
>
> Ideally, when $\text{Inversion}=\text{Sampler}^{-1}$, $\mathcal{E}^{rec}$ is currently at its minimum value of 0. **Uni-Inv is trying to be such a inverse of the given sampler**, so it is related to the sampler. Our derived local error $O(\Delta t\_i^3)$ serves $\mathcal{E}^{rec}$ rather than $\mathcal{E}^{dis}$.
>
> For high-order solvers, even though we know $\mathcal{E}^{dis} = O(\Delta t\_i^3)$, **the reconstruction errors  $\Vert Z\_0 - \text{Solver}(\text{Solver}^{inv}(Z\_0)) \Vert$ of these solvers are usually unknowed**, thus leading to unpredictable results.
>
> ---
>
> > **[W3] Can you explain appendix section D. Why the inversion method should be related to samplers? The purpose of inversion and inversion based editing is to find the noise given an existing real world image and we are not suppose to know how the image is generated.**
> >
>
> Thanks for your question. Continuing from the previous response, the design objective of Uni-Inv is:
>
> $$
> \min\_\text{Inversion} \mathcal{E}^{rec} = \min\_\text{Inversion} \Vert Z\_0 - \text{Sampler}(\text{Inversion}(Z\_0)) \Vert.
> $$
>
> Intuitively, an optimal solution is:
>
> $$
> \text{Inversion}=\text{Sampler}^{-1}.
> $$
>
> Therefore, although Uni-Inv is not such an optimal solution, **it also has the design goal of minimizing reconstruction error, so its specific implementation inevitably depend on the adopted sampler**.
>
> BTW, the core innovation of Uni-Inv is to propose the idea of "take a step back and take a step forward", which makes each inversion step similar to corresponding denoising step to approach better inversion methods.
>
> ---

---

> > ### Comment · Reviewer_xe1o · 2025-11-24
> > **My apology for the ambiguous comments**
> >
> > Thanks for the rebuttals. My concerns on W3-W6 are addressed.
> > However, I would like to make myself clearer on W1 and W2. I totally understand Heun is a solver, but it can also be used for inversion by simply taking inverse timesteps. If you use Heun for inversion, the difference between Heun and your proposed method is only on Line 3 in the "for" loop (Algorithm 1). i.g. change the current v_i to (v_i + v_i-1)/2.
> >
> > So why v_i is better than (v_i+v_i-1)/2 ?
> >
> > BTW, does the "Heun" in Table 1 refer to the same thing I mentioned above?

---

> > > ### Author Response · Authors · 2025-11-24
> > > **Response to Reviewer xe1o's comments about Heun and Uni-Inv**
> > >
> > > Thank you for your recognition and thoughtful consideration of our response. We also apologize for not having an exact understanding of your comments. Your insights about Heun is correct, Heun can be directly used as an inversion for the Euler sampler. If we represent our proposed Uni-Inv step as:
> > >
> > > $$
> > > \text{Euler}:~ Z\_{t\_{i-1}}=Z\_{t\_i}+(t\_{i-1}-t\_i){v\_{i}}, \\
> > > \text{Uni-Inv}:~ \widehat{Z}\_{t\_i}=\widehat{Z}\_{t\_{i-1}} - (t\_{i-1}-t\_i)\widehat{v}\_i.
> > > $$
> > >
> > > Then what we understand as the method you are referring to is (we name it Heun-Inv):
> > >
> > > $$
> > > \text{Euler}:~ Z\_{t\_{i-1}}=Z\_{t\_i}+(t\_{i-1}-t\_i){v\_{i}}, \\
> > > \text{Heun-Inv}:~ \widehat{Z}\_{t\_i}=\widehat{Z}\_{t\_{i-1}} - (t\_{i-1}-t\_i)\frac{\widehat{v}\_{i-1} + \widehat{v}\_i}{2},
> > > $$
> > >
> > > which we think is equivalent to changing the current $v\_i$ to $(v\_i + v\_{i-1})/2$ in Alg. 1, Line 3. Here we compare the inversion and reconstruction effectiveness of “Heun-Inv” on the first 500 samples of CC3M validation set (the entire dataset evaluation consumes too much time, so we only utilize the first 500 samples for rebuttal) using SD3:
> > >
> > > | **Method** | **Uncond. MSE**$^\downarrow\_{10^3}$ | **Uncond. PSNR**$\uparrow$ | **Uncond. SSIM**$^\uparrow\_{10^2}$ | **Uncond. LPIPS**$^\downarrow\_{10^2}$ | **Cond. MSE**$^\downarrow\_{10^3}$ | **Cond. PSNR**$\uparrow$ | **Cond. SSIM**$^\uparrow\_{10^2}$ | **Cond. LPIPS**$^\downarrow\_{10^2}$ |
> > > | --- | --- | --- | --- | --- | --- | --- | --- | --- |
> > > | Euler | 30.87 | 16.31 | 57.76 | 30.15 | 28.45 | 16.74 | 57.01 | 30.46 |
> > > | Heun | 25.83 | 16.85 | 67.19 | 26.72 | 26.71 | 16.82 | 64.09 | 27.85 |
> > > | **Heun-Inv** | 16.28 | 20.42 | 70.85 | 20.81 | 9.27 | 22.77 | 78.88 | 12.64 |
> > > | **Ours** | 11.94 | 21.49 | 78.33 | 13.31 | 8.05 | 23.11 | 81.89 | 9.76 |
> > >
> > > It shows that **Heun-Inv is superior to typical solvers, but still inferior to the proposed Uni-Inv**. To theoretical analyze the reconstruction error of Heun-Inv, we first rewrite Heun-Inv step as:
> > >
> > > $$
> > > \widehat{Z}\_{t\_i}=\widehat{Z}\_{t\_{i-1}} - (t\_{i-1}-t\_i)\frac{\widehat{v}\_{i-1} + \widehat{v}\_i}{2}=\frac{\widehat{Z}\_{t\_{i-1}} - (t\_{i-1}-t\_i)\widehat{v}\_{i-1}}{2} + \frac{\widehat{Z}\_{t\_{i-1}} - (t\_{i-1}-t\_i)\widehat{v}\_{i}}{2},
> > > $$
> > >
> > > which can be seen as the average of Euler inversion and Uni-Inv from the perspective of the local step. The local reconstruction error of Uni-Inv is $O(\Delta t\_i^3)$, while the local reconstruction error of Euler inversion is:
> > >
> > > $$
> > > (t\_{i-1}-t\_i) \Vert v\_\theta(Z\_{i-1}, t\_{i-1}) - v\_\theta(Z\_i,t\_i) \Vert \le (t\_{i-1}-t\_i) ( \Vert Z\_{i-1} - Z\_i \Vert + \Vert t\_{i-1}-t\_i \Vert ) = O(\Delta t^2).
> > > $$
> > >
> > > Therefore, adopting Heun as the inversion can improve accuracy, but still retain $O(\Delta t\_i^2)$ term in the reconstruction error, which is theoretically and experimentally worse than Uni-Inv.
> > >
> > > Additionally, **the "Heun" we reported in Tab. 1 does not refer to the “Heun-Inv” method, it simply refers to the original Heun inversion and sampling**. Similar to RF-Solver and FireFlow, here we utilize the Heun method with the reversed timesteps as the inversion method, and directly use the original Heun sampler for reconstruction. Therefore, the Heun method we compared in Tab. 1 is similar to other solvers in that it reduces discretization errors but does not directly focus on reconstruction errors, so it does not perform particularly well in Tab. 1.

---

> > > > ### Comment · Reviewer_xe1o · 2025-11-25
> > > > **Thanks for the experiments**
> > > >
> > > > The new experiments addressed my concerns. Therefore I would like to increase my score to 6.

---

> > > > > ### Author Response · Authors · 2025-11-25
> > > > > **Response to Reviewer xe1o's feedback**
> > > > >
> > > > > It is our pleasure to have in-depth discussion with you regarding the method design and theoretical foundation of the manuscript. Thank you for your thoughtful consideration and appreciation.

---

> ### Author Response · Authors · 2025-11-21
> **Response to Reviewer xe1o (2)**
>
> > **[W4] Continue from 3, what is the data you used in Table 1? Are they generated or real world data? Are they generated with a certain type of solver?**
> >
>
> Thank you for your attention to the experiment details. For **Tab. 1**, we use the Conceptual Captions validation dataset (CC3M, [https://huggingface.co/datasets/pixparse/cc3m-wds](https://huggingface.co/datasets/pixparse/cc3m-wds)). As claimed in their page, **the images and their raw descriptions are harvested from the web. These images are generally real and not generated through models by us**. This type of real-world data is also more difficult to accurately invert and reconstruct, resulting in baselines’ poor performance. **Fig. 6, H.8, H.9** are constructed using this dataset.
>
> ---
>
> > **[W5] Some of the claims are questionable. For example, the trajectory of flow models is not straight, only the conditional trajectory is straight.**
> >
>
> Thank you for pointing it out. In experiments, the trajectory of flow is also generally not straight. However, although we have mentioned the property of trajectory’s straightness in our manuscript, **the proposed method does not rely on this property**, as can be seen from the experiments based on diffusion in the appendix **Sec. D.2, D.3, E**. The trajectory of diffusion is definitely non-straight, but it still benefits from Uni-Inv and Uni-Edit in reconstruction and editing tasks.
>
> In fact, mentioning these theoretical properties of flow models in the manuscript is more to indicate that **these properties have given us great inspiration** in the design of the proposed method. For example, in Uni-Inv, thanks to the concise formulation of rectified flow we can think out the design of "take a step back and take a step forward", while the first-order approximation is also inspired by the trajectory’s straightness often mentioned in flow-related papers.
>
> ---
>
> > **[W6] The authors may consider to discuss some related to works for inversion in flow based models. For example [1][2]**
> >
>
> Thank you for the introduction of relevant works. These cutting-edge works are very inspiring. We have discussed FlowEdit [1] in the sampling-based methods part of the related works and have cited RFDS [2] at related works of the manuscript after studying it.
>
> > FlowEdit [1] provides an inversion-free editing method, which is in line with our goal of expending diffusion-based applications to flow models through the design of a pure sampling method. The optimization-based inversion method proposed by RFDS [2] theoretically analyzes the properties of the flow model and considers it as a prior, which is similar to the base of our inversion-based editing method. Such a prior can be extended to diverse applications through plug and play modules. We have further discussed about this kind of expansion in our subsequent response to reviewer 7YEx's [W2].
> >
>
> [1] FlowEdit: Inversion-Free Text-Based Editing Using Pre-Trained Flow Models.
> [2] Text-to-Image Rectified Flow as Plug-and-Play Priors.
>
> ---

---

### Official Review · Reviewer_aUpR · 2025-11-01

**Soundness:** 3
**Presentation:** 4
**Contribution:** 3
**Rating:** 6
**Confidence:** 3

**Summary:**

This submission proposes novel inversion and editing methods specifically designed for flow-based models. For inversion, the paper introduces Uni-Inv, which incorporates an additional correction procedure prior to the inversion step. For editing, leveraging delayed injection, the paper presents Uni-Edit, featuring a predictor-corrector mechanism and a velocity fusion step to enable effective edits while preserving regions irrelevant to the desired changes. The key contributions of this work are the introduction of an extra correction procedure in both inversion and editing processes, as well as the use of velocity fusion in the editing step.

**Strengths:**

1. The paper introduces a predictor-corrector procedure to enhance performance in both inversion and editing tasks. This approach is original and has been extensively evaluated. The method employs delayed injection, which is a widely used technique in image editing and is not an original contribution of this work. The velocity fusion technique, adapted from prior research on region-aware editing, is incorporated into the proposed method. Overall, the paper achieves a satisfactory level of originality and has the potential to inspire future research.

2. The submission presents comprehensive and well-designed experiments, particularly in the Appendix, demonstrating the advantages of the proposed method over baseline approaches in both inversion and editing. Various backbone models and conditioning scenarios are considered and evaluated. In addition, the authors extend their method to video editing and diffusion models, showcasing its flexibility and generalization capabilities.

3. The paper is clearly written and well organized. Figures such as Figure 3 and Figure 5 help readers better understand the algorithm through effective demonstrations and diagrams. Overall, the readability is excellent.

**Weaknesses:**

1. The main concern lies in the ablation study of key components, which is crucial for verifying their effectiveness. The submission presents an ablation study in Table C.1. For unedited regions, metrics such as PSNR and SSIM appear reliable for measuring the accuracy of background preservation. However, regarding the edited regions, relying solely on CLIP similarity may be insufficient to fully demonstrate editing performance. Including human evaluation or visual results in the ablation study would provide a more comprehensive assessment.

2. The implementation of “w/o Uni-Inv” is unclear. To effectively ablate Uni-Inv, experiments similar to those in Figure 4 with different velocity settings could be conducted in the evaluation.

3. Additionally, velocity fusion does not appear to have a significant impact. Its removal results in only a slight decrease in PSNR and SSIM, with CLIP similarity remaining almost unchanged.

**Questions:**

1. What is the difference between “m^{in Corr.}_i = 1” and “w/o Corr.” in the ablation study presented in Table C.1? Both configurations appear to represent Uni-Edit without Correction, yet they yield significantly different results in PSNR/SSIM and CLIP similarity. How can this discrepancy be explained?

2. A more detailed analysis of these key components, supported by visual examples, should be provided to better elucidate the ablation study results. Specifically, it would be helpful to understand why certain metrics increase or decrease when specific components are removed.

---

> ### Author Response · Authors · 2025-11-21
> **Response to Reviewer aUpR (1)**
>
> Dear Reviewer aUpR,
>
> Thanks for your appreciation of the originality, effectiveness, and readability of our work. Your suggestions on ablation studies are very helpful. Here is our response regarding this matter.
>
> ---
>
> > **[W1] The main concern lies in the ablation study of key components, which is crucial for verifying their effectiveness. The submission presents an ablation study in Table C.1. For unedited regions, metrics such as PSNR and SSIM appear reliable for measuring the accuracy of background preservation. However, regarding the edited regions, relying solely on CLIP similarity may be insufficient to fully demonstrate editing performance. Including human evaluation or visual results in the ablation study would provide a more comprehensive assessment.**
> >
>
> Thank you for your suggestions on experimental design and result presentation. Based on your suggestions, **we have added qualitative analyses in the latter half of Sec. C.1 of the revised appendix, and we have also presented the visual results in Fig. C.3.** In visual comparison, to compensate for the insufficient explanatory power of the metrics, we have emphasized and discussed background details (which is difficult to capture by PSNR or SSIM) and visual editing effects (such as whether the result is overexposure) to provide more comprehensive evaluations and analyses.
>
> ---
>
> > **[W2] The implementation of “w/o Uni-Inv” is unclear. To effectively ablate Uni-Inv, experiments similar to those in Figure 4 with different velocity settings could be conducted in the evaluation.**
> >
>
> Thank you for your insightful advice. **The reported “w/o Uni-Inv” corresponds to $\boldsymbol{v}(\cdot,t\_{i-1})$ in Fig. 4. We further added the results of using $\boldsymbol{v}(\cdot,t\_{i})$ for inversion in editing tasks**, as shown in the table below.
>
> | **Method** | **Structure Distance**$^\downarrow\_{10^3}$ | **Background PSNR**$\uparrow$ | **Background SSIM**$^\uparrow\_{10^2}$ | **CLIP Whole**$\uparrow$ | **CLIP Edited**$\uparrow$ |
> | --- | --- | --- | --- | --- | --- |
> | Inv. $\boldsymbol{v}(\cdot,t\_{i-1})$ (i.e., w/o Uni-Inv) | 40.87 | 21.93 | 74.90 | 25.54 | 21.93 |
> | Inv. $\boldsymbol{v}(\cdot,t\_{i})$ | 36.95 | 23.82 | 80.25 | 25.99 | 22.08 |
> | **Ours** | 21.40 | 24.96 | 86.11 | 26.39 | 22.72 |
>
> Empirically, **the more accurate the inversion, the more favorable it is for preserving the background for image editing and avoiding generation collapse**. Meanwhile, if the inverted noisy latent maintains a good structure for reconstruction, it also helps editing to accurately and stably edit appropriate regions, thus improving editing results. The results in the table demonstrate this viewpoint and also reflect the significance of Uni-Inv for applications such as image editing.
>
> Furthermore, **we have added results and analyses of the above content in Tab. C.1 and Sec. C.1 of the appendix and presented visual comparisons in Fig. C.3 (c)**, which further reflects the impact of different inversions on editing.
>
> ---

---

> ### Author Response · Authors · 2025-11-21
> **Response to Reviewer aUpR (2)**
>
> > **[W3] Additionally, velocity fusion does not appear to have a significant impact. Its removal results in only a slight decrease in PSNR and SSIM, with CLIP similarity remaining almost unchanged.**
> >
>
> Thank you for your comment. In fact, **the phenomenon you observed aligns with our argument that naive delayed injection has little effect on flow models**, as our argument reflects that velocity alone cannot play a sufficiently effective and reliable role in image editing tasks. This is also the reason why we propose the correction steps.
>
> **However, the role of velocity fusion is more of a smooth complement to correction**. From the visualization in **Fig. 5**, it can be seen that the correction step significantly eliminates concepts of conflicts with the editing objectives, however, it can also be seen that this elimination is too strong and coarse-grained in the sampling early steps. Therefore, the velocity used to move the sample to the next timestep needs to be more cautious, while avoiding editing-irrelevant regions being changed and editing-related regions not being fully edited. Under such goals, **velocity fusion is the most reasonable choice in our proposed framework**.
>
> To validate this point, we have first compared the impact of different optional velocities on editing metrics, as shown in the table below. $\boldsymbol{m}\_i^\text{in V.F.}=\boldsymbol{1}$ is equal to $\boldsymbol{v}^F=\boldsymbol{v}^T$, and $\boldsymbol{m}\_i^\text{in V.F.}=\boldsymbol{0}$ is the same as  $\boldsymbol{v}^F=\boldsymbol{v}^S$. From the perspective of the trade-off of background preservation and editing effects, **velocity fusion achieves a "inflection point" effect**, while other choices lead to the degradation of one ability more obvious than the improvement of the other.
>
> | **Method** | **Structure Distance**$^\downarrow\_{10^3}$ | **Background PSNR**$\uparrow$ | **Background SSIM**$^\uparrow\_{10^2}$ | **CLIP Whole**$\uparrow$ | **CLIP Edited**$\uparrow$ |
> | --- | --- | --- | --- | --- | --- |
> | $\boldsymbol{m}\_i^\text{in V.F.}=\boldsymbol{1}$ | 22.92 | 24.62 | 85.50 | 26.39 | 22.74 |
> | $\boldsymbol{m}\_i^\text{in V.F.}=\boldsymbol{0}$ | 21.78 | 25.01 | 86.13 | 26.22 | 22.57 |
> | **Ours** | 21.40 | 24.96 | 86.11 | 26.39 | 22.72 |
>
> Not only above, **we have also added visual comparisons of the velocity selection in Fig. C.3 (b) and discussed the comparison results in Sec. C.1 in the appendix**. It is worth noting that the visualization results indicate that although velocity fusion has a less significant impact on the overall performance image editing, **it clearly helps a lot to control the quality of editing details**. It also reflects that velocity fusion is the preferred choice for making image editing more stable and accurate.
>
> ---
>
> > **[Q1] What is the difference between “m^{in Corr.}\_i = 1” and “w/o Corr.” in the ablation study presented in Table C.1? Both configurations appear to represent Uni-Edit without Correction, yet they yield significantly different results in PSNR/SSIM and CLIP similarity. How can this discrepancy be explained?**
> >
>
> Thanks for pointing out the ambiguity in the manuscript. Given $v\_i^- = v\_i^T - v\_i^S$, our proposed correction step in Uni-Edit is $s\_i=\omega(t\_{i-1}-t\_i)(1+m\_i)\odot v\_i^-$. Then, “$m^\text{in~Corr.}\_i=1$” indicates the correction is changed to $s\_i=\omega(t\_{i-1}-t\_i)(1+1)\odot v\_i^-$, and “$\text{w/o Corr.}$” represents the correction is changed to $s\_i=0$. Therefore, the former eliminates the region-adaptive guidance of the correction step, while the latter eliminates the correction step itself.
>
> The reason for the significant difference is that **the former has changed from only correcting editing-related regions to applying correction to the entire image**, directly causing the background to lose its preservation from the velocity obtained from source condition, sacrificing background preservation for more obvious editing effects. **The latter represents eliminating the correction step in Uni-Edit**, and only relies on the fused velocity for editing, which will strongly weaken the editing effect.
>
> In the revised manuscript, **we have provided visual comparisons between these two conditions and our proposed method in Fig. C.3 (d), and discussed it in Sec. C.1. Eliminating the correction can make editing fail, while the correction without region-adaptive guidance can easily damage the background and even lead to overexposure results**. These analyses not only demonstrate the effectiveness of the correction step in Uni-Edit, but also demonstrate the significance of region-adaptive guidance for accuracy and controllability of image editing.
>
> ---

---

> ### Author Response · Authors · 2025-11-21
> **Response to Reviewer aUpR (3)**
>
> > **[Q2] A more detailed analysis of these key components, supported by visual examples, should be provided to better elucidate the ablation study results. Specifically, it would be helpful to understand why certain metrics increase or decrease when specific components are removed.**
> >
>
> Thanks for your kind suggestions. **We have added visual comparisons and detailed analyses in Sec. C.1 of the manuscript** to help understand the roles of these components. In terms of analysis conclusions we can draw that:
>
> - The more favorable the inversion is for reconstruction, the more helpful it is for image editing. At the same time, Uni-Inv provides SOTA inversion results for flow models, making the background of the edited results closer to the source image, while the foreground not collapse.
> - The correction step in Uni-Edit is the main driving force behind the successful editing of delayed injection paradigms on flow models.
> - Region-adaptive guidance helps to better determine the editing-related regions during the correction process, avoiding undesired background damage or image distortion caused by excessive correction.
> - Although velocity fusion is not the main driving force of editing, it helps foreground editing and background preservation of corrected samples, while also providing smooth control over the quality of details and the effectiveness of detail editing.
>
> ---

---

### Official Review · Reviewer_J9uw · 2025-11-01

**Soundness:** 3
**Presentation:** 3
**Contribution:** 3
**Rating:** 6
**Confidence:** 4

**Summary:**

The paper introduces UniEdit-Flow, a training-free, model-agnostic pipeline for flow models that tackles (i) exact inversion problem that aims to transitions from a data to the corresponding latent via a predictor–corrector scheme (Uni-Inv) that aligns an implicit-Euler step and proves an bound for error of the proposed method, , and (ii) editing method (Uni-Edit) using region-adaptive and delayed injection for locality preservation while changing the target subject.

**Strengths:**

– Mostly well written and easy to follow.

– Problem and solution are well-motivated; the design is simple yet effective.

– Impressive results: consistent gains for inversion (Table 1) and editing (Table 2), with strong qualitative examples on SD3 and FLUX.

– Demonstrations across varied applications suggest good generality and a principled approach.

**Weaknesses:**

– The transition from §4.2 to §4.3 is hard to track. In Fig. 5 and the surrounding text, a correction step uses $v_{i}^{S}$ to move the latent to a higher-noise state, then $v_{i}^{T}$ to correct under the new prompt; however, it’s unclear which velocity is applied next—the phrase “apply the current editing velocity to move the latent to $\tilde Z_{t_{i-1}}$ is ambiguous.

– Fig. 5 introduces $v_{i}^{F}$ without prior definition; its relation to $v_{i}^{S}$ and $v_{i}^{T}$, and its role in updating $\tilde Z_{t_{i-1}}$, are not explained at that point in the main text. I would prefer an explicit introduction in the text rather than only in the caption (as in Fig. 3).

**Questions:**

– Please define $v_{i}^{F}$ when it first appears and clarify how it
is constructed from $v_{i}^{S}$ and $v_{i}^{T}$ (e.g., fusion rule,
weights, timing), or at least indicate what it is and where it will
be specified, rather than only in the Fig. 3 caption.

– Specify precisely what “current editing velocity” refers to at the step that updates $\tilde Z_{t_{i-1}}$ and provide the explicit update equation.

---

> ### Author Response · Authors · 2025-11-21
> **Response to Reviewer J9uw**
>
> Dear Reviewer J9uw,
>
> We sincerely appreciate your recognition of our manuscript and proposed methods. The questions and suggestions you raised about the writing and expression are to the point. We are glad to discuss how to modify this parts with you here. We have divided your review into multiple independent questions and answered them separately.
>
> ---
>
> > **[Q1] The transition from §4.2 to §4.3 is hard to track.**
> >
>
> Thank you for pointing the logical issue in the manuscript. **Sec. 4.2** presents the core idea of Uni-Edit, which is to empower image editing through the correction step. Subsequently, based on this idea, **Sec. 4.3** adds fine-grained region control to the implementation of this idea, addressing the background preservation issues that need to be specificly noted. **We have added transitional sentences at the beginning of Sec. 4.3 to avoid hard logical tracking**:
>
> *To further precisely correct concepts that need to be edited while avoiding excessive damage to the background, a simple idea is to use a mask to determine the edit-relevant regions.*
>
> ---
>
> > **[Q2] In Fig. 5 and the surrounding text, a correction step uses $v\_i^S$ to move the latent to a higher-noise state, then $v\_i^T$ to correct under the new prompt; however, it’s unclear which velocity is applied next—the phrase “apply the current editing velocity to move the latent to $\tilde{Z}\_{t\_{i-1}}$ is ambiguous.
> Specify precisely what “current editing velocity” refers to at the step that updates $\tilde{Z}\_{t\_{i-1}}$ and provide the explicit update equation.**
> >
>
> Thank you for pointing the ambiguity in the manuscript. The “current editing velocity” means the velocity utilized for sample denoising in the editing process, which is *$\boldsymbol{v}\_i^F$* of Uni-Edit. To make the expression more clear, **we have refined the last sentence of Sec. 4.2 and replaced the pharse “current editing velocity” with *$\boldsymbol{v}\_i^F$* itself**:
>
> *Then, we apply $\boldsymbol{v}\_i^F$ (introduced in next part) to move from $\boldsymbol{\check{Z}}\_{t\_i}$ to $\boldsymbol{\tilde{Z}}\_{t\_{i-1}}$, and finally achieving a proper edit.*
>
> Meanwhile, the equation of sample denoising has also been added at the indicated location in **Sec. 4.3**:
>
> *Then the sample is updated by $\boldsymbol{\tilde{Z}}\_{t\_{i-1}} = \boldsymbol{\check{Z}}\_{t\_i} + (t\_{i-1} - t\_{i}) \boldsymbol{v}^F\_i$.*
>
> ---
>
> > **[Q3] Fig. 5 introduces $v\_i^F$ without prior definition; its relation to $v\_i^S$ and $v\_i^T$, and its role in updating $\tilde{Z}\_{t\_{i-1}}$, are not explained at that point in the main text. I would prefer an explicit introduction in the text rather than only in the caption (as in Fig. 3).
> Please define $v\_i^F$ when it first appears and clarify how it is constructed from $v\_i^S$ and $v\_i^T$ (e.g., fusion rule, weights, timing), or at least indicate what it is and where it will be specified, rather than only in the Fig. 3 caption.**
> >
>
> Thank you for your kind advice. **We have added specific expressions for clearer expression of the method in Sec. 4.3**:
>
> *For the subsequent sample update, we fuse the target and source velocities using $\boldsymbol{m}\_i$ and $(\boldsymbol{1} - \boldsymbol{m}\_i)$ as respective weights, thus forming the velocity fusion: $\boldsymbol{v}\_i^F=\boldsymbol{m}\_i \odot \boldsymbol{v}^T\_i + (\boldsymbol{1} - \boldsymbol{m}\_i) \odot \boldsymbol{v}^S\_i$. Then the sample is updated by $\boldsymbol{\tilde{Z}}\_{t\_{i-1}} = \boldsymbol{\check{Z}}\_{t\_i} + (t\_{i-1} - t\_{i}) \boldsymbol{v}^F\_i$.*
>
> ---

---

### Author Response · Authors · 2025-11-21
**General Response**

Dear reviewers and AC,

We sincerely appreciate your valuable time and effort spent reviewing our manuscript.

As reviewers highlighted, our manuscript is well-written with great readability (`J9uw`, `aUpR`, `xe1o`, `7YEx`), proposing a simple yet effective (`J9uw`) method with satisfactory originality (`aUpR`), generality (`J9uw`, `aUpR`), and superior experimental results (`J9uw`, `aUpR`, `xe1o`, `7YEx`).

Moreover, we really appreciate your constructive feedback on our manuscript, which provides many impressive insights and helpful suggestions.

In response to the comments, we have carefully revised and enhanced our manuscript as well as crafting the review response. The modified parts in the updated paper are highlighted in $\textcolor{green}{\bf{green}}$. The specific issues discussed include:

- Detailed expression of proposed method (Reviewer `J9uw`).
- Visual comparisons, detailed equations, and analyses of ablation studies (Reviewer `aUpR`).
- Detailed methodology and motivation (Reviewer `xe1o`).
- Method’s flexibility and generalization for diverse applications, as well as experiments introducing image conditions (Reviewer `7YEx`).

Sincerely,

The Authors

---

### Author Response · Authors · 2025-12-03
**General Response for Rebuttal Summarizing**

Dear reviewers and AC,

We sincerely thank you for your conscientious attitude, in-depth thinking, time and attention during the review of our manuscript. The following is a rebuttal summary of the questions with our corresponding responses.

---
| **Reviewer** | **Rating (and feedback)** | **Strengths** | **Questions** | **Our response** |
| --- | --- | --- | --- | --- |
| `J9uw` | 6 | Well-written and well-motivated. Simple yet effective. Impressive results. Principled approach with good generality. | **1.** Transition from §4.2 to §4.3 is hard to track. **2.** Text defination of "current velocity" and $v\_i^F$. | **1.** We add transitional sentences at the beginning of Sec. 4.3. **2.** We add definations of used symbols in the main text. |
| `aUpR` | 6 | Satisfactorily original and extensively evaluated. Flexible and generalizable. Excellent presented and well organized. | **1.** Visual results of ablation studies. **2.** Implementation clearification about inversion method in editing ablations. **3.** Impact of velocity fusion appears to insignificant. | **1.** We add qualitative analyses of ablations in Sec. C.1 and Fig. C.3. **2.** We provide additional experiments using various inversion velocity presented in Fig. 4. **3.** We compare the visual results of velocity fusion's ablations, showing its significance for the quality and editing effect of image details. |
| `xe1o` | 4 (concerns addressed, increase score to 6) | Well-written. Quite good results. | **1.** The difference between Uni-Inv and high-order solvers and their different local error analysis. **2.** Why the inversion method should be related to samplers. **3.** Data type, questionable claims, and related work discussions. | **1.** We formulaicly present the differences of motivation and error analysis between Uni-Inv and high-order solvers, and demonstrate through theory and experiments that Uni-Inv is better than directly using Heun for inversion. **2.** We explain the reasons that Uni-Inv is related to samplers through the formulaic language. **3.** We show that our evaluations are conducted using real images and our method is not dependent on straight flow trajectory claims, while providing discussions of more related works. |
| `7YEx` | 6 (keep positive rating towards acceptance) | Reasonable logic and good writing. Strong performance. Justified theoretical analysis. | **1.** Hyper-parameter selection in the evaluation. **2.** Applications on image-conditioned editing. **3.** Utilization of attention maps. | **1.** We indicate that we have fixed the hyper-parameters across different images to ensure the comparison is fair. **2.** We additionally provide image-conditioned editing results in Sec. G. **3.** We claim that attention maps can be used in our framework, while we would like our method to be robust to the architectural change. |
---

Sincerely,

The Authors

---

### Meta-Review · Area_Chair_mfuD · 2025-12-10

**Summary:**

All four reviewers find the paper clearly written and empirically strong, with a simple but effective predictor–corrector inversion (Uni-Inv) and a practical editing scheme (Uni-Edit) that works well across several flow and diffusion models. The main hesitation was whether Uni-Inv is really new compared to standard high order solvers like Heun, and whether the editing pipeline is more than a careful recombination of known ideas. After the rebuttal and extra experiments, my view is that the paper offers a solid, timely contribution to inversion and editing for flow models, and it merits acceptance.

**Reviewer Concerns:**

xe1o: Concerns about Uni-Inv being essentially Heun, unclear benefit over other high order methods, unclear link to samplers, data type in Table 1, and some questionable claims and missing related work. These were addressed with a clearer distinction between discretization vs reconstruction error, new experiments with a Heun based inversion (Heun-Inv), clarification that Table 1 uses real CC3M images, and added related work. Reviewer explicitly stated their concerns were resolved.

aUpR: Wanted stronger ablations and clearer implementation of "w/o Uni-Inv" and velocity fusion. Authors added extra quantitative and qualitative ablations, clarified the different variants, and explained the role of velocity fusion as a stabilizing component. These concerns are largely addressed, though the gain from velocity fusion remains moderate numerically, which the authors acknowledge and explain.

J9uw: Asked for clearer notation, definitions, and transition between Sections 4.2 and 4.3. These are straightforward writing issues that the authors fixed.

7YEx: Raised hyperparameter sensitivity, potential image conditioned extensions, and the possibility of using attention maps. The authors clarified that hyperparameters are fixed per benchmark (not per image) and showed image conditioned applications via IP-Adapter and related plugins. They chose not to rely on attention to stay architecture agnostic, but also explained that such modules can be integrated on top. The main concerns are reasonably addressed and do not seem blocking.

**Reviewer Scores:**

J9uw: Stayed at 6. With the clarifications, I expect them to keep this score.

aUpR: Stayed at 6. Given the expanded ablations and explanations, I expect them to keep a positive 6.

xe1o: Went from 4 to 6 after rebuttal and new experiments. I agree with this final score.

7YEx: Stayed at 6, explicitly reaffirming a positive rating.

Overall, after discussion all reviewers converge at 6, consistent with an accept decision.

---

### Decision · Program_Chairs · 2026-01-26

Accept (Poster)